# Functionally constrained human proteins are less prone to mutational instability from single amino acid substitutions

Maryam May [1,4], Aaron Chuah [1,3,4], Nicole Lehmann[1], Llewelyn Goodall [1,2], Vicky Cho [1] & T. Daniel Andrews [1,2] ✉

Missense mutations that disrupt protein structural stability are a common pathogenic mechanism in human genetic disease. Here, we quantify potential disruption of protein stability due to amino acid substitution and show that functionally constrained proteins are less susceptible to large mutational changes in stability. Mechanistically, this relates to greater intrinsic disorder among constrained proteins and to increased B-factors in the ordered regions of constrained proteins. This phenomenon means that constrained proteins exhibit smaller stability effects due to missense mutations, and partly explains why overtransmission of pathogenic missense variation is less prevalent in genetic disorders characterised by protein truncations. We show that the most functionally constrained proteins are depleted of both destabilising and overly-stabilising amino acid variation in disease-free populations. Despite this, amino acid substitutions with large stability effects in functionally constrained proteins are still highly prevalent among pathogenic human genetic variation. Importantly, we observe that there are approximately five times more missense variants with large stability effects than there are unambiguous loss-of-function mutations. Missense variants with disruption of stability effects recapitulate the per-gene patterns of functional constraint observed with protein truncating loss-of-function variation, yet their relative abundance abrogates difficulties encountered when estimating functional constraint for the shortest human genes.

The pathogenicity effects of missense mutations that result in altered protein stability have been widely accepted for some time[1]. Due to evolutionary trade-offs, most proteins are only marginally stable, yet must function in the busy physiochemical environment of the cell, mixed among proteases that degrade unfolded proteins[2]. Hence, missense mutations that disrupt stability of the encoded protein have potentially large functional consequences. And a large proportion of all possible missense variants have functionally relevant stability effects on the encoded protein[3,4]. The functional effects of these protein mutants range from interfering with correct protein folding to disruption of protein-protein interactions, to affecting cellular localization of the protein[5]. Destabilization of a protein structure underlies the pathogenic mechanism of many and diverse missense mutations that cause genetic disease[6].

Missense mutations with protein stability effects are common - and may be the pathogenic mechanism of a significant minority, or even the majority, of missense substitutions[1]. Mutations that disrupt protein stability can result in complete loss of protein function, but

[1]The John Curtin School of Medical Research, The Australian National University, Canberra, Australia. [2]School of Computing, College of Engineering, Computer Science and Cybernetics, The Australian National University, Canberra, Australia. [3]Present address: National Cancer Centre Singapore, Singapore, Singapore. [4]These authors contributed equally: Maryam May, Aaron Chuah. ✉e-mail: Dan.Andrews@anu.edu.au

may often simply reduce (hypomorphic mutations) or increase activity (hypermorphic and gain-of-function mutations) from the wild-type level, and in some cases alter the activity of the protein (neomorphic mutations)[7]. Furthermore, at a systems level, the functional impact and/or penetrance of these mutations is dependent also on gene functional constraint, potential haploinsufficiency and the inheritance pattern of the mutant phenotype (including dominance/recessiveness and/or polygenicity)[8,9]. Given these diverse effects of mutations, and that these effects are not readily distinguished at the sequence level, in the context of this study we have chosen a broader term for mutants that affect protein stability as Disruption-of-Stability (DoS) mutants. Here, this term, DoS mutant, makes no assumptions about mechanism, inheritance or penetrance, yet simply indicates that the mutant protein is functionally altered (or disrupted) compared to the wild-type protein, especially if this has deleterious consequences that may lead to the variation becoming subject to purifying selection.

Protein stability is commonly measured as the change in the Gibbs free energy ($\Delta G$) between the native, folded state of a protein and the unfolded state. Hence, the change in stability due to an amino acid substitution can be quantified as the change in $\Delta G$ value between the wild-type and variant protein – represented as $\Delta\Delta G$. Protein instability due to mutation may reduce the energetic difference between the folded and unfolded states, thus destabilizing the protein. Or an amino acid substitution may increase the rigidity of the protein, thereby overly stabilizing the protein. Much empirical data has been collected to characterize the relationship between amino acid substitutions and structural stability[10]. This data has been used to train a diversity of accurate computational tools to predict the $\Delta\Delta G$ due to single and/or multiple amino acid substitutions[11], and these may be applied to predict stability changes due to genetic variation at a proteome-wide level.

With human genome sequences collected from increasingly larger numbers of individuals, it has been observed that genes subject to high functional constraint do not display the expected numbers of detrimental variants that are observed among functionally redundant genes[12]. A gene subject to strong functional constraint will harbor many fewer protein Loss-of-Function (pLoF) variants than a functionally redundant gene. To quantitate functional constraint, the 'loss-of-function observed/expected upper bound fraction' (LOEUF) score and the relative rank of the LOUEF score among all genes[12] (with genes commonly partitioned bins by decile) has been produced from the genomic variation in the GnomAD dataset[12]. Briefly, the LOEUF value represents the 90% confidence interval of the ratio of observed and expected pLoF variation for a given gene, where the expected pLoF variation is calculated from empirical trinucleotide mutation frequencies and the composition of these in each given gene[12]. The LOEUF scores calculated from population-scale genome datasets count only the most unambiguous of potential pLoF mutants: nonsense, frameshift and canonical splice donor-acceptor mutants[13]. However, these pLoF mutants are comparatively rare. Estimating functional constraint from these requires very large datasets of genomes, from more than one hundred thousand individuals[12]. And even then, the shortest genes in the genome may contain very few or no pLoF mutations by chance and remain underpowered for detection of functional constraint[14]. To address this lack of resolution inherent in unambigious pLoF variation, especially in shorter genes, an approach utilizing $S_{het}$ scores has recently been proposed[14]. This employs a population genetics model with machine learning, using gene features derived from auxiliary datasets that are predictive of pLoF constraint, including gene expression information[14]. The inclusion of this auxiliary data better predicts functional constraint for the shortest 25% of human genes, through pooling of information from better-powered genes that are subject to similar constraint[14]. This approach has been implemented the GeneBayes framework[14] that allows the model to be further extended with iterative inclusion of additional data.

We have previously made exhaustive predictions of the stability effects of all possible amino acid substitutions within all human proteins (www.stabilitysort.org)[15]. Here we investigate the relative depletion of missense variants with high predicted protein stability impacts from genes under strong functional constraint. With this, we identify thresholds at which missense mutations disrupt protein stability and become DoS variants. From these we identify that the most functionally constrained genes in the genome are comparatively less prone to mutational instability than functionally redundant proteins. Also employing these thresholds to count DoS variation, we calculate LOEUF scores using these more frequent DoS mutations and show this recapitulates known constraint patterns.

## Results

### The tolerated range of protein stability effects due to missense variation

We computed predictions of protein stability change due to mutation ($\Delta\Delta G$) for all possible single amino acid substitutions in every human protein. We used the MAESTRO tool[16] to predict the stability effect, measured as $\Delta\Delta G$, of all single amino acid substitutions using structural models predicted by AlphaFold2[17]. We chose the MAESTRO tool because of its low computational cost and good predictive performance[11,18]. Furthermore, MAESTRO is available as standalone software, thereby facilitating computation in a parallel, high-performance computing environment. With these exhaustive stability change predictions we were able to observe the stability variation of variant sets of differing functional types. Figure 1a shows that the set of all possible changes for every residue in every protein ranged from highly destabilizing $\Delta\Delta G$ values above 2.5 kcal/mol to overly stabilizing changes with values below −1.5 kcal/mol. Figure 1a also compares the observed, tolerated stability variation for each of the functional constraint categories (LOEUF deciles) estimated for missense variants from the GnomAD database[12], and stability variation observed among all observed pathogenic, missense mutations from the ClinVar database[19]. As might be expected, the range of $\Delta\Delta G$ values for missense variants observed in the most functionally constrained LOEUF decile (bin 0) is narrower than the range of pathogenic missense variants recorded in ClinVar. The range of tolerated DoS becomes increasingly broad with decreasing functional constraint. Yet, the range of $\Delta\Delta G$ values for the least constrained genes (bin 9) is still narrower than that exhibited by all possible mutants (Fig. 1a). Noteworthy is the skew in ClinVar pathogenic variation towards destabilizing variation (higher $\Delta\Delta G$), compared with population variation observed in the GnomAD dataset.

For comparison, Fig. 1b shows an additional functional constraint metric, $S_{het}$ that has robust performance with shorter genes[14]. As observed among proteins of decreasing LOEUF, the range of tolerated stability variation becomes narrower with increasing selective pressure, though the differentiation of this effect is less pronounced between these categories than for LOEUF decile bins.

### Instability thresholds for missense disruption-of-function variants

We were motivated to identify instability thresholds at which the most functionally constrained human proteins no longer harbored presumed-pathogenic destabilizing or stabilizing amino acid variation in population-scale datasets. With this means to classify missense variation as disrupting function due to protein instability, we may label missense variation that is likely DoS variation. Our presumption was that among these constrained proteins, variants with large stability effects that affect protein function will become subject to purifying selection and hence be absent from the sequence record of disease-free human populations. And among proteins subject to less functional constraint, variants of similar stability effects will be variably present due to lower purifying selection active in the encoding genes.

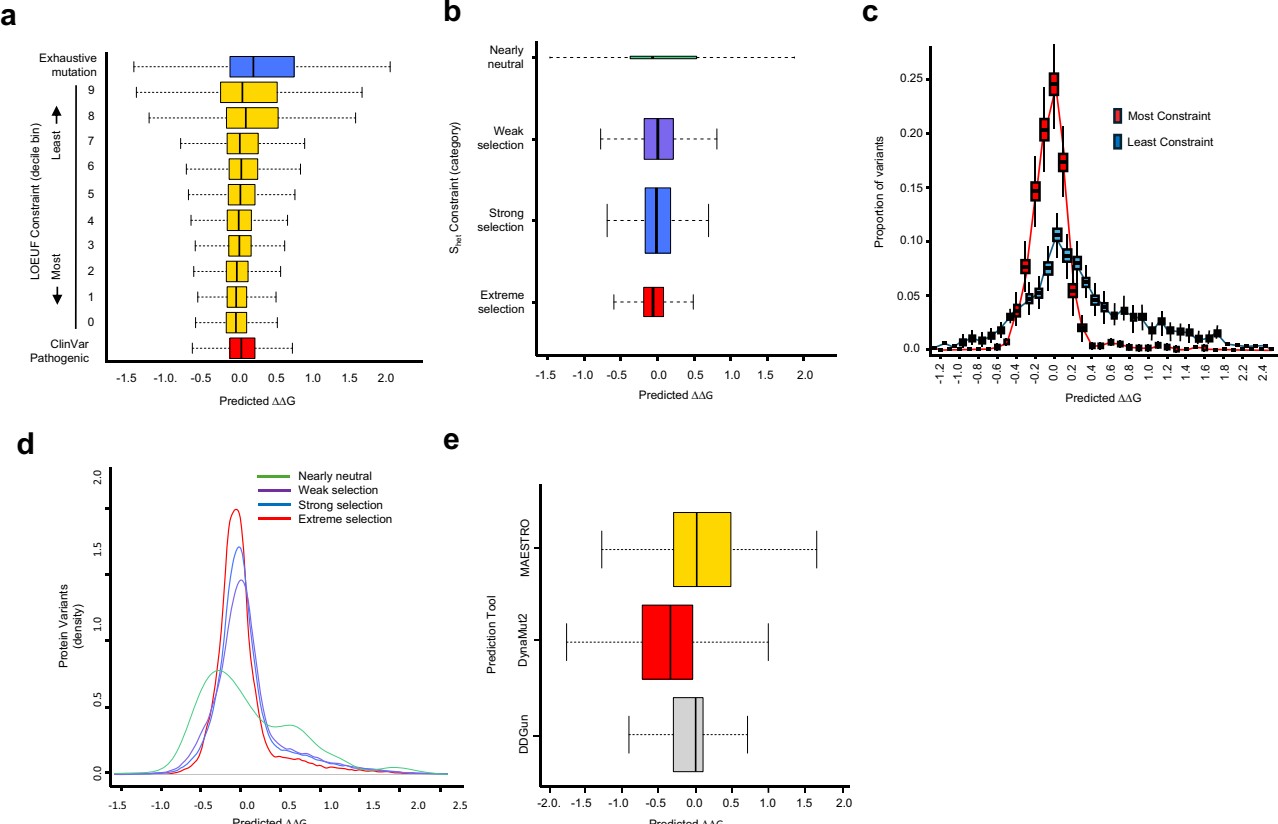

**Fig. 1 | Functional constraint and predicted protein instability due to single amino acid substitutions.** For each boxplot, the medians are marked by a central bar in bold black, bounds of each box are the interquartile range (IQR) and whiskers extend 1.5*IQR from each box. **a** Range of predicted protein instability (ΔΔG) for missense variants of defined functional importance. Variant categories are ClinVar Pathogenic (all missense change from the ClinVar database with the annotation CLNSIG=Pathogenic; $n = 27515$), Loss-of-Function Observed/Expected Upper Bound Fraction (LOEUF) Constraint 0-9 (GnomAD population variation stratified by gene functional constraint LOEUF deciles; bin 0 $n = 297505$, bin 1 $n = 179407$, bin 2 $n = 156475$, bin 3 $n = 127725$, bin 4 $n = 124798$, bin 5 $n = 137012$, bin 6 $n = 121523$, bin 7 $n = 112796$, bin 8 $n = 85660$, bin 9 $n = 70802$), and Exhaustive mutation (comprehensive prediction of stability effects for all possible single amino acid substitutions

in all human proteins; $n = 3744146$). **b** Predicted protein instability variation grouped by gene and stratified by $S_{het}$ selection constraint categories (extreme $n = 34866$, strong $n = 173464$, weak $n = 67513$, neutral $n = 254$). **c** Comparative depletion of single amino acid variation by predicted ΔΔG between LOEUF constraint categories: most constrained (red; LOUEF bin 0) and least constrained (blue; LOUEF bin 9) in a disease-free cohort (1000 Genomes, Phase 3 release, $n = 3115$ individuals for each constraint category). **d** Comparative depletion of single amino acid variation by predicted ΔΔG between $S_{het}$ selective constraint categories (green: nearly neutral, purple: weak selection, blue: strong selection, red: extreme selection). **e** Range of ΔΔG values predicted by three widely used tools (MAESTRO, DynaMut2 and DDGun) on the same input set of amino acid substitutions ($n = 2403$). Source data are provided as a Source Data file.

To identify stability change thresholds at which missense variation becomes subject to purifying selection, we sought the maximum and minimum values of ΔΔG that were tolerated within the most constrained human proteins (LOUEF bin 0). Figure 1c shows the variation in the predicted ΔΔG of amino acid substitutions observed in a disease-free human population of 3115 individuals (1000 Genomes, Phase 3 release)[20], where observed missense variation in the most- and least-constrained LOUEF decile bins (0 and 9, respectively) is plotted separately. The variation observed among the most constrained human proteins (LOEUF bin 0) was presumed to be almost entirely benign and that from the least constrained proteins (LOEUF bin 9) was assumed to contain a considerable number that are deleterious changes. Figure 1c shows the expected excess of destabilizing and overly-stabilising variation present among LOEUF bin 9 proteins as longer tails extending to higher and lower ΔΔG values. Using this observed depletion of protein instability variation in the most constrained (bin 0) genes, we chose upper and lower ΔΔG cut-offs of 0.5 and −0.5 kcal/mol, respectively. These thresholds represent the instability boundaries beyond which a missense variant will become subject to purifying selection in a functionally constrained gene. We repeated this threshold-finding analysis with the selection pressure

categories defined by $S_{het}$ value[14]. The depletion of missense variation from the 'extreme selection' gene category was compared to the 'nearly neutral' category (Fig. 1d) and this comparison supports the thresholds chosen above.

Using these −0.5 < ΔΔG < 0.5 thresholds, we calculate a false-discovery rate (FDR) of 6%, for DoS mutants being annotated as benign variants in ClinVar (CLNSIG=Benign). With this same benign variant set, we also obtained comparable FDR values with AlphaMissense pathogenicity scores[21] (4.3% of benign variants called as 'pathogenic') and substantially higher FDR values than with PolyPhen2[22] (14.9% called 'probably damaging') or SIFT[23] (29.9% called 'deleterious'). ΔΔG values alone with these thresholds for DoS variation (−0.5 < ΔΔG < 0.5) have good performance for detecting missense variation that may cause a disrupted molecular phenotype.

These ΔΔG threshold values (−0.5 < ΔΔG < 0.5) are narrow compared to those empirically measured[24,25]. For comparison, we replicated ΔΔG predictions for 2419 amino acid substitutions observed across eleven proteins with two additional tools: DynaMut2[26] and DDGun[27]. Both additional tools have been shown to provide similar accuracy to MAESTRO[11]. The predicted values of these additional tools also show the narrower range of predicted ΔΔG values than that

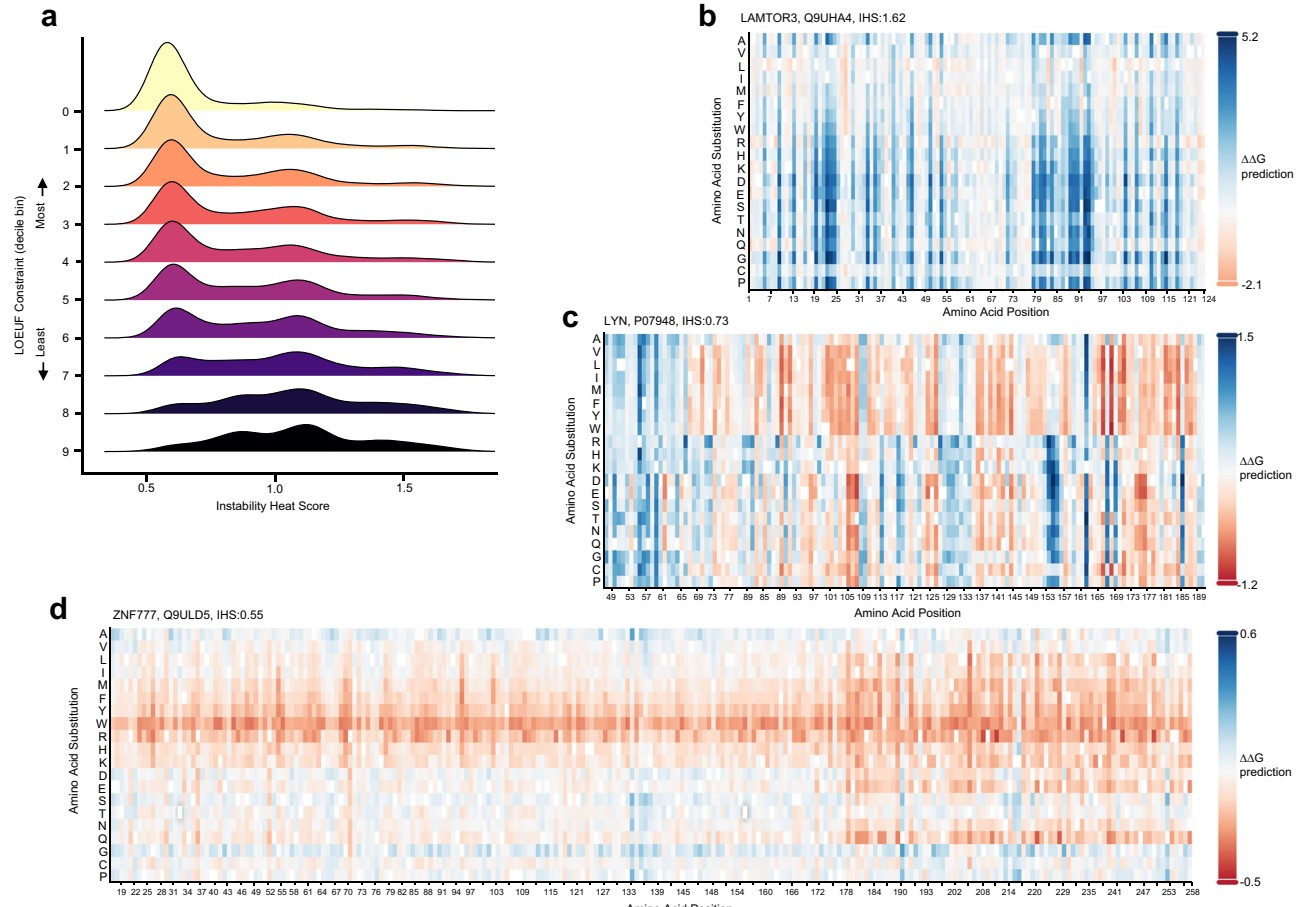

**Fig. 2 | Instability Heat Scores (IHS). a** Distribution of IHS by Loss-of-Function Observed/Expected Upper Bound Fraction (LOEUF) functional constraint category (partitioned by LOEUF decile). Representative heat maps of IHS (from www.stabilitysort.org) for (**b**) LAMTOR3 (Q9UHA4, IHS:1.62, pLI:0.0, LOEUF:0.548, LOEUF decile:2), **c** LYN (P07948, IHS:0.73, pLI:1.0, LOEUF:0.221, LOEUF decile:0) and (**d**) ZNF777 (Q9ULD5, IHS:0.55, pLI:1.0, LOEUF:0.035, LOEUF decile:0) genes. See the key for each plot for the range of instability values in each heat map. Blue denotes destabilising substitutions, red denotes stabilising substitutions and white denotes the wild-type amino acid at that position. Source data for (**a**) is provided as a Source Data file.

observed in empirical data (Fig. 1e), lending support to the finding above that at relatively minor predicted instability values, there are observable depletions of missense variants among functionally constrained genes compared to those that are relatively unconstrained.

With the instability thresholds above, 18.4% of all ClinVar pathogenic missense variants (5115/27766 = 0.184; CLNSIG=Pathogenic) are classified as DoS variations, that lie outside of these boundaries for tolerated stability of functionally benign mutations.

Supplementary Fig. 1a and b show that both LOEUF and $S_{het}$ functional constraint categories, respectively, are heterogeneous with respect to gene lengths, with longer genes being associated with higher constraint. Supplementary Fig. 1c shows that tolerated levels of mutational instability vary somewhat with protein length, especially among the shortest proteins with fewer than 400 amino acids.

**Proteins subject to strong functional constraint exhibit muted mutational instability**

We developed an aggregate protein metric, the Instability Heat Score (IHS), to quantitate the susceptibility to mutational stability effects of any given protein, subject to single amino acid substitution mutations. For any given protein, the IHS is the average of the absolute stability change ($|\Delta\Delta G|$) of all possible substitutions that result in a predicted stability change that lies outside the thresholds determined above ($-0.5\,\text{kcal/mol} < \Delta\Delta G < 0.5\,\text{kcal/mol}$). This metric is built on the thresholds determined in the previous section, yet represents the

susceptibility (or fragility) of a protein to instability introduced by mutations that substitute single amino acids. Proteins with higher IHS are fragile to mutational instability, whereas those with low IHS are 'colder' and, in general, do not exhibit as pronounced mutational instability effects. IHS varies considerably between proteins and the distribution of values differ with relative category of functional constraint (Fig. 2a). Figure 2b–d show representative heatmaps of proteins that are either more fragile (high IHS) or are colder (low IHS) to amino acid substitution, from a stability perspective. We have generated heatmaps of IHS of all human proteins (available at www.stabilitysort.org). In general, and as expected, proteins are rarely uniformly prone to mutational instability along their length.

We observed that when proteins are partitioned into constraint decile bins by LOEUF that more constrained proteins have broadly lower IHS and therefore are generally less prone to large changes in stability due to missense mutations (Fig. 3a). When we included only proteins that have well-powered LOEUF estimates[28] (see Supplementary Data 6 from Chen et al.[28]; GnomAD v4.1), the differentiation of IHS with functional constraint is slightly decreased (Fig. 3b), yet the overall trend towards generally lower IHS in constrained proteins remains. Supplementary Fig. 1d shows this same trend in a similar boxplot of IHS calculated using variable instability thresholds, dependent on protein length, which partially normalises for the protein length variation observed between constraint categories (Supplementary Fig. 1a). Seeking a potential mechanism for this trend, we considered the

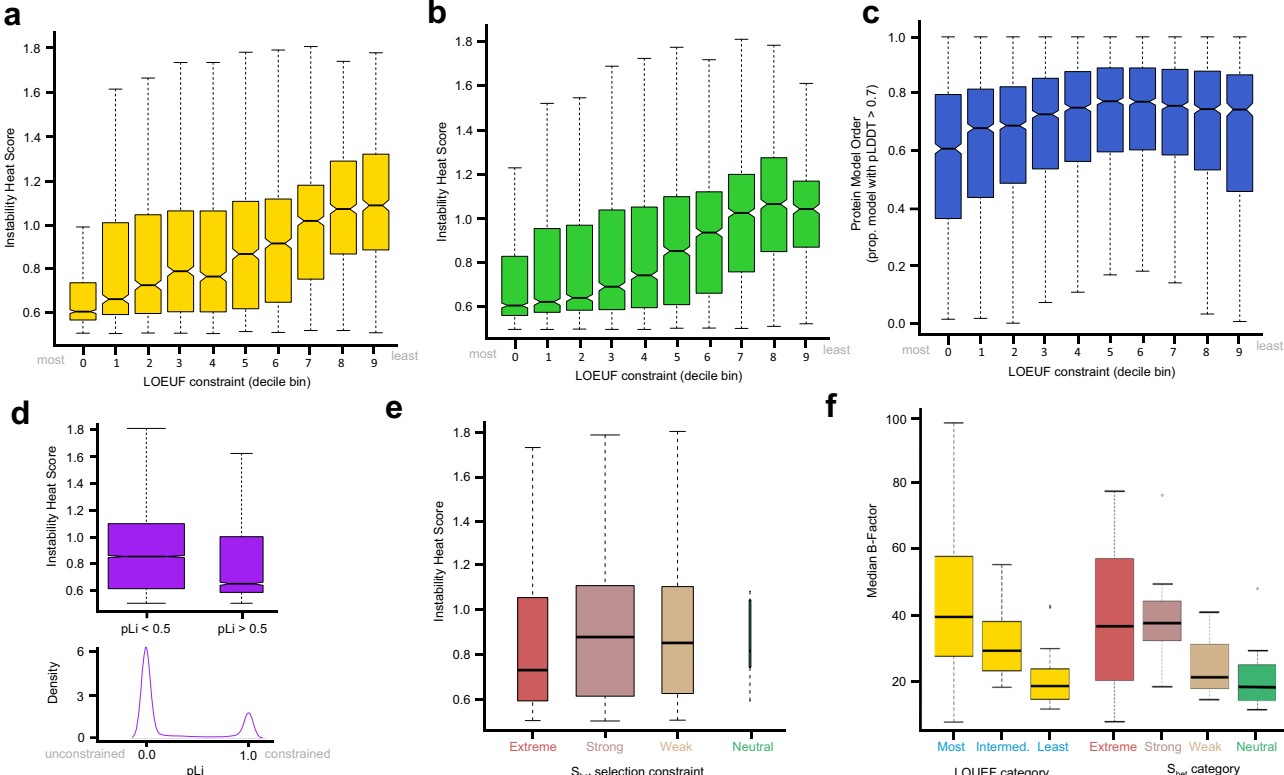

**Fig. 3 | Protein robustness to stability effects of amino acid change with respect to functional constraint category.** In all boxplots, the central bar in each box represents the median value of each respective category, the bounds of each box are the interquartile range (IQR), whiskers extend 1.5*IQR from each box and the central notches (where present) represent an approximation of the 95% confidence interval of the median box value. **a** Observed range of Instability Heat Score (IHS) for all human proteins, grouped by Loss-of-Function Observed/Expected Upper Bound Fraction (LOEUF) functional constraint decile bins (bin 0 is most constrained and bin 9 is least constrained; bin 0 $n = 1031$, bin 1 $n = 1035$, bin 2 $n = 1049$, bin 3 $n = 1083$, bin 4 $n = 1071$, bin 5 $n = 1053$, bin 6 $n = 1079$, bin 8 $n = 972$, bin 9 $n = 736$), (**b**) observed range of IHS by LOEUF decile bins for all well-powered human proteins (bin 0 $n = 1557$, bin 1 $n = 1601$, bin 2 $n = 11553$, bin 3 $n = 1505$, bin 4 $n = 1473$, bin 5 $n = 1479$, bin 6 $n = 1454$, bin 8 $n = 1210$, bin 9 $n = 803$), (**c**) proportion of protein (AlphaFold2-model) with order greater than pLDDT > 0.7 by LOEUF

functional constraint decile bin (bin 0 $n = 671$, bin 1 $n = 716$, bin 2 $n = 795$, bin 3 $n = 834$, bin 4 $n = 901$, bin 5 $n = 864$, bin 6 $n = 913$, bin 8 $n = 778$, bin 9 $n = 555$), (**d**) comparative range of IHS for well-powered human genes (defined from Chen et al.[28]; $n = 13218$) grouped by low- and high-pLi scores, with respective low ($n = 10017$) and high ($n = 3201$) functional constraint. The width of boxes in the upper plot indicates the relative number of genes present in each category, and (**e**) observed range of IHS for all human genes partitioned by $S_{het}$ selection constraint categories (Extreme $n = 1747$, Strong $n = 6485$, Weak $n = 1901$, Neutral $n = 7$). Box widths are proportional to the number of genes included in each category, and (**f**) median B-factor for experimentally solved protein structures, grouped by LOEUF and $S_{het}$ constraint categories. LOEUF constraint decile categories are grouped by Most (bins 0–2; $n = 43$), Intermediate (bins 3–6; $n = 23$) and Least (bins 7–9; $n = 18$). Shet categories are grouped by Extreme ($n = 34$), Strong ($n = 19$), Weak ($n = 12$) and Neutral ($n = 18$). Source data are provided as a Source Data file.

contribution of relative protein disorder with IHS. Figure 3c shows a decrease in structural order (increased intrinsic disorder) quantified by the percentage of the AlphaFold2 model for each structure with pLDDT greater than 70%. Visually, this mirrors lower IHS in constrained protein LOEUF deciles groups.

Quantities such as pLi[29] provide an alternative metric with which to assess the relationship of IHS with functional constraint[30]. The pLi score observed for all human genes has a strongly bimodal distribution[30] as shown in the lower distribution in Fig. 3d, with a greater number of genes displaying low functional constraint, as expected[30]. We partitioned genes by low (pLi < 0.5) and high (pLi > 0.5) functional constraint and observed the range of IHS for each category (upper boxplot in Fig. 3d). Similar to LOEUF decile grouping, the two pLi groups indicate that the higher constraint genes (pLi > 0.5) have a general property of lower IHS, as indicated by the non-overlapping notches between the two boxes. This is despite the two categories containing a very large number of genes each, potential diluting any subtle signals present in this information.

Similarly, we replicated this analysis using the gene selection pressure categories defined by $S_{het}$[14]. Figure 3e shows boxplots of IHS by $S_{het}$ selection constraint categories (extreme, strong, weak and

neutral selection) following the naming scheme of Zeng et al.[14]. As with the pLi comparison above, the number of genes present in each selection constraint category varies, and there are very few genes that are neutral. However, the median IHS for genes subject to extreme selection is lower than the strong and weak categories, though the small number of genes in the neutral category yields an inconclusive comparison with other categories.

The broad phenomenon that the most functionally constrained proteins may be less prone to instability by some metrics is recapitulated in the median B-score values of proteins with experimentally solved structures (Fig. 3f, Supplementary Table 1). Noteworthy to this investigation is that more than half of the experimentally solved structures are partial structures (see Supplementary Table 1) and in general do not include intrinsically disordered regions. Proteins from the most functionally constrained categories have higher median B-score values than those of the least constrained categories. These proteins in solved structures have greater uncertainty (and thus greater movement) of the positions of atoms in the protein crystals. This same trend is also present among gene categories (strong, weak and neutral selection) partitioned by $S_{het}$ score (Fig. 3f), though B-factors for proteins in the extreme selection show the broadest

variation and encompasses that of all of the other three categories. Supplementary Fig. 1e, f recapitulate the same trend of increasing B-factor with rising functional constraint, but also indicate that this also co-varies with increased protein length. This is congruent with the general decrease in the magnitude of instability from amino acid substitution observed with increasing protein length in Supplementary Fig. 1c. As already pointed out, very many of these experimental structures are partial structures, the trend towards elevated B-factor in these partial structures of larger proteins indicates this is an inherant property of larger proteins, rather than a technical artefact of crystallisation of larger proteins.

We note from Fig. 3a that the most constrained proteins (bin 0) have a median IHS of just 0.61. But despite this, the greatest number of pathogenic DoS variants present in ClinVar (CLNSIG=Pathogenic) are also found among these most functionally-constrained proteins, and of these, 20.4% are amino acid substitutions with functionally disruptive stability effects (at threshold −0.5 < ΔΔG < 0.5).

### Instability Heat Scores and relative pathogenic effect of missense variation

We explored the relevance of protein IHS in the context of missense variation observed in a large phenotype-associated genome sequence dataset, obtained from individuals with autism and their families[31]. This dataset aggregates genomic variation from 63237 individuals and calculates Bayes Factor (BF) values to quantify evidence of phenotypic association from multiple variant classes (protein truncating variation (PTV), missense variation, indels and copy number variation) and modes of inheritance[31]. We used BF values to investigate covariation with both IHS and predicted ΔΔG scores, at both a per gene and per variant level. The original work with BF values showed that PTV contributes most evidence that implicates the majority of the 72 ASD-linked genes[31]. But intriguingly, for a minority of genes, missense

variation provides more than 90% of the evidence that links them to an autism phenotype in proband individuals (eg. *PTEN, SLC6A1, DYNC1H1, DEAF1, AP2S1*)[31]. The 72 ASD-linked genes are almost all from the most functionally constrained classes (54 of 72 genes are from LOEUF decile bin 0 - meaning the top 10% of most constrained genes). The 72 ASD-linked genes are slightly skewed towards lower IHS (65.5% of ASD-linked genes have an IHS of less than 0.6, compared with 51.0% among all genes). Only three of the 72 ASD-linked genes (*DEAF1, PTEN, AP2S1*) harbor potential DoS variation is ASD probands, yet this variation represents 83.0% of all de novo missense variation in these 72 ASD-linked genes in proband individuals, compared with 15.5% for the set of all genes (Fig. 4a, b; see accompanying Source Data file). Potential DoS variation in only two ASD-linked genes (*PTEN* and *AP2S1*) contributes most to this differential between gene sets. These genes are outliers compared to the other ASD-linked genes, having the highest two IHS of the 72 ASD-associated genes, and are two of the least functionally constrained proteins in this set (Fig. 4c). These metrics (BF, IHS and ΔΔG) used together seem a useful means of further segregating genes that may contribute to phenotype through missense variation. The Source Data files accompanying Fig. 4a, b provide further information of IHS, LOEUF functional constraint and de novo missense mutations observed outside DoS thresholds in probands. Figure 4c shows BF values for PTV and missense variation (MisB and MisA classes, after nomenclature of Fu et al.[31]) plotted against IHS. This comparison shows that *PTEN* and *AP2S1* have the highest susceptibility to DoS mutations of the 72 ASD-linked genes, and this is reflected in the unusually high contribution of missense variation (BF values from MisB) to detecting this phenotypic association.

### Comparison of constraint between pLoF and DoS variation

More point mutations will cause missense variation than will cause nonsense or canonical splicing changes. Hence, DoS variation has the

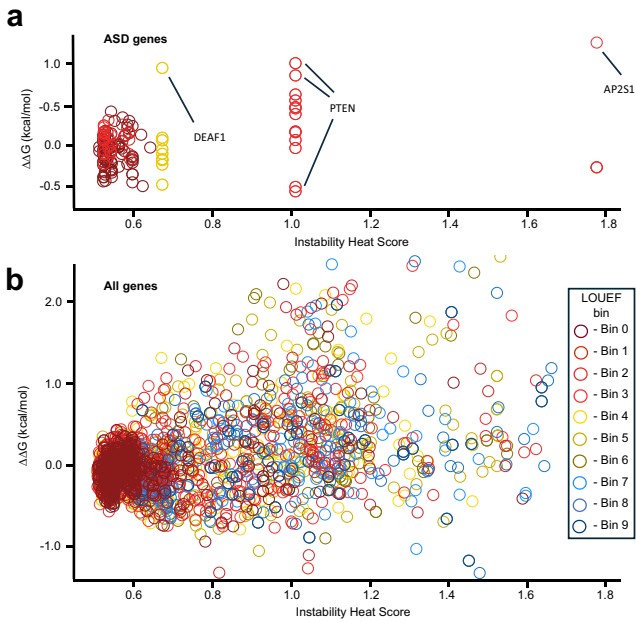

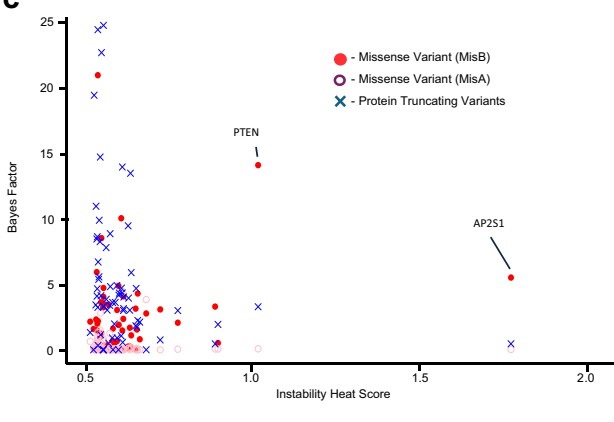

**Fig. 4 | Genes linked to phenotypes in people with autism are robust to potential protein instability effects of de novo missense variation. a** Almost all ASD-linked genes[31] are highly functionally constrained (points colored by Loss-of-Function Observed/Expected Upper Bound Fraction (LOEUF) bin), have lower Instability Heat Score (IHS) and little observed de novo genetic variation has stability effects outside the −0.5 < ΔΔG < 0.5 thresholds that imply functional importance. Whereas (**b**) among all observed de novo genetic variation present in the same individuals there is a broad range of functional constraint, IHS and stability effects on the encoded proteins, including much that is likely to compromise

function. In both (**a**, **b**), the same symbols are used, as denoted in the key to (**b**). **c** With Bayes Factor (BF) values for 72 ASD-linked genes, for each of three variant classes (using BF values and nomenclature from Fu et al.[31]), two proteins (PTEN and AP2S1) display both higher IHS and MisB BF values, and additionally, contain amino acid substitutions in ASD probands with ΔΔG values that imply function DoS variation of these proteins. Green X denote protein truncating variation, purple circles denote MisA variation and red dots denote MisB variation. Source data are provided as a Source Data file.

potential to greatly outnumber protein LoF (pLoF) variation, which has been the gold-standard variation type used to estimate gene functional constraint. Using observed variation from the GnomAD dataset[12], and the DoS thresholds chosen above ($-0.5 < \Delta\Delta G < 0.5$), there are five times more observed DoS variants per gene (median = 35) than pLoF variants (median = 7).

We used the set of observed DoS variants per gene from the GnomAD dataset (v4.1) and calculated the expected number of DoS variants[32] to calculate LOEUF scores per human gene (see Materials and Methods). Figure 5a shows that comparison between LOEUF scores calculated with the same methodology, but with either pLoF and DoS variation as input. The range of LOEUF scores calculated from DoS variants is narrower than that calculated with pLoF variants, with few genes having values less than 0.5 or greater than 1.5. When genes are partitioned by LOEUF decile values, Fig. 5b shows that the greatest

similarity of the gene sets present in each LOEUF decile bin (calculated from both DoS and protein pLoF variant classes) is among the most constrained genes. This conservation of gene sets is less similar among the least constrained functional constraint decile bins, where pLoF LOEUF will perform less well for shorter genes.

Summary data provided in a Zenodo deposition (https://doi.org/10.5281/zenodo.14873427) contain a reference dataset of observed and expected variant counts for both pLoF and DoS variation and the LOEUF scores calculated from these for all human genes (EnsEMBL GRCh38 p14) with canonical transcripts having a one-to-one mapping with UniProt identifiers for which an AlphaFold2-predicted structural model exists. Genes in the summary data have also been annotated with the raw difference in LOUEF scores calculated by pLoF and DoS variation.

Table 1 shows selected genes from the external summary data (https://doi.org/10.5281/zenodo.14873427) with the largest differences in functional constraint between pLoF and DoS variants. Of note, *HLA-B* and *HLA-DRB1* genes have LOUEF scores calculated with DoS variants that are more than double that calculated with pLoF variation. These genes encode MHC Class I and MHC Class II proteins and their high polymorphism is related to their immune function. This functional polymorphism and does not include LoF polymorphism. Many of the genes in Table 1 with the most disparate LOEUF scores are among the shorter genes in the human genome and have few, if any, observed pLoF variants in the GnomAD dataset. Other genes have very few DoS variants, while having relatively larger numbers of clear pLoF variants. Table 1 includes two gene examples of functionally important roles for the gene described and include OMIM identifiers for disease phenotypes that result from pLoF mutations in these genes.

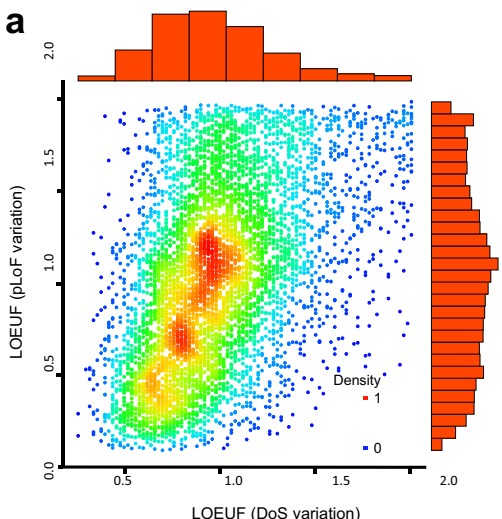

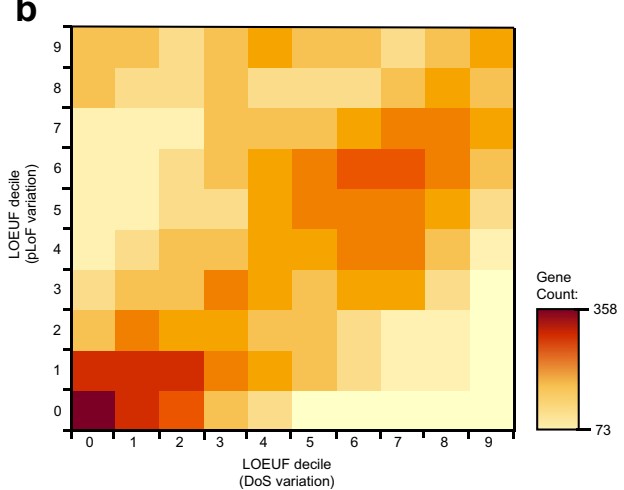

**Fig. 5 | Comparison of gene-specific Loss-of-Function Observed/Expected Upper Bound Fraction (LOEUF) scores derived from LoF and DoS variant sets.** **a** Density plot of the comparison of LOEUF scores between those calculated with pLoF and DoS variation, where the colours from blue to red represent the density of points for individual genes by pLoF and DoS LOEUF measures, and histograms in red to the top and right show the distribution of values of LOUEF for DoS and pLoF variation, respectively, and (**b**) a heatmap of observed gene counts by LOEUF decile bin, allowing comparison of gene membership of each grouping by LOEUF scores, calculated from pLoF and DoS variation. Source data are provided as a Source Data file.

## Discussion

From an exhaustive computational mutagenesis experiment and the predictions of the resulting changes to protein stability, we have observed that in aggregate the most functionally constrained human proteins have muted changes to overall protein stability due to the effects of single amino acid substitutions. That some proteins are structurally more capable of accommodating amino acid substitutions has been the subject of long interest (see Tokuriki and Tawfik[4]). Both the higher intrinsic disorder that typifies constrained, haploinsufficient genes[33] and the higher network centrality of constrained genes[34,35] go some way to explaining this trend. The size distribution of proteins versus their functional constraint has also received recent attention, and we observe here that longer proteins generally have smaller stability effects due to mutation than shorter proteins. This is despite our understanding that proteins with fewer buried residues are also more robust to mutational instability[4,36]. It is important to consider this in the context that the often-lethal effect of mutations to essential, constrained proteins have a subsequent high cost to organismal fitness. These effects may have led to selective pressure driving these proteins towards relatively fewer contacts between residues and therefore increased inherent flexibility to amino acid substitution. This is reflected in the higher median B-factors we observed from experimentally solved structures in the most functionally constrained proteins. It is striking how interwoven this property (captured by IHS) is with the overall functional constraint to which a given gene/protein is subject. One important consequence of this will be that fewer missense variants in strongly constrained disease genes will result in pathogenic effects, due to a central pathogenic mechanism of missense mutation being muted in these genes. Hence, when searching affected cohorts and/or personal genomes for causal mutations, if the phenotype or condition results from alteration or ablation of the function of highly constrained genes, the contribution of missense variation may be slight, and it is important to prioritise other mutation types in these genes.

**Table 1 | Selected genes with greatest LOEUF difference between values calculated with either DoS or pLoF variation**

| Gene | Length (nt) | Transcript | Missense DoF | | | GnomAD pLoF | | | OMIM Phenotype-Genotype (MIM numbers) |
|---|---|---|---|---|---|---|---|---|---|
| | | | Observed Variants | Expected Varaints | LOEUF | Observed Variants | Expected Variants | LOEUF | |
| HLA genes - high missense variants, antigenic diversity | | | | | | | | | |
| HLA-DRB1 | 798 | ENST00000360004 | 78 | 31.00 | 2 | 6 | 10.24 | 1.16 | |
| HLA-B | 1086 | ENST00000412585 | 128 | 60.05 | 2 | 10 | 18.59 | 0.91 | |
| Appear constrained in GnomAD, due to very few observed LoF | | | | | | | | | |
| RGS1 | 627 | ENST00000367459 | 43 | 29.25 | 1.86 | 1 | 9.39 | 0.51 | |
| AQP9 | 885 | ENST00000219919 | 139 | 81.23 | 1.94 | 3 | 11.72 | 0.66 | |
| RAD23B | 1227 | ENST00000358015 | 84 | 67.79 | 1.48 | 1 | 20.92 | 0.23 | |
| PRPS1L1 | 954 | ENST00000506618 | 103 | 29.87 | 2 | 2 | 8.01 | 0.79 | |
| LAMTOR3 | 372 | ENST00000499666 | 31 | 23.53 | 1.76 | 1 | 8.66 | 0.55 | |
| Appear unconstrained in GnomAD, but with penetrant mutant phenotypes | | | | | | | | | |
| KLLN | 534 | ENST00000445946 | 22 | 52.18 | 0.6 | 3 | 2.62 | 1.89 | 615107, 612105 |
| SDHAF1 | 345 | ENST00000378887 | 28 | 66.18 | 0.58 | 1 | 0.91 | 1.90 | 619166, 612848 |

While keeping the preceding points in mind, functionally constrained proteins are not impervious to mutation-induced instability. We find that the most functionally constrained proteins disproportionately contain more observed pathogenic variants in the ClinVar database, potentially due to their high functional importance and potential to produce penetrant disease phenotypes. We have previously produced a tool, StabilitySort (www.stabilitysort.org)[15], to prioritise potentially pathogenic missense mutations using the population variation of protein stability to represent the 'unusualness' of the stability effects of a given variant in a given protein. This has proven a remarkably useful means to prioritise potentially causal genetic variation[15]. Strikingly, the distribution of pathogenic missense variation recorded in ClinVar is skewed towards destabilizing change, though our data also indicate that stabilizing structural change is just as subject to purifying selection. This underrepresentation of stabilizing pathogenic variation may represent some combination of i) a recording bias, ii) a difference in pathogenic mechanism, or iii) that the disease phenotypes due to this form of variation are milder.

Among genetic disorders that result from disruption of highly functionally constrained genes, our understanding of the contribution of missense variation remains poor, despite these being one of the more predominant forms of observed genetic variation. Here we describe the comparative susceptibility of constrained proteins to the stability effects of amino acid substitutions. In an analysis of the observed de novo missense variation identified in people with autism and their families, we note the predominance of PTV in highly functionally constrained genes (LOEUF bins 0 and 1), yet point out the trend towards both higher IHS, and a greater contribution of missense variation to BF values, among slightly less functionally constrained genes (pLoF LOEUF decile 2 and greater). The prevalent assumption in the field has long been that ambiguities of the functional effects of missense variants obscures true functional signal. Here, we have better characterized one aspect of these ambiguities faced in the interpretation of the functional effect of missense genetic variation.

Sequence conservation metrics are at the core of many tools for the prediction of the functional importance of missense mutations[21–23,37]. Genes and gene regions are variably tolerant of missense variation, and much attention has been directed towards the predictive usefulness of relative depletion of missense variation to identify functionally constrained elements. In linear coding sequences, the regional depletion of missense variation combined with an amino acid substitution score and PolyPhen2 scores has provided the joint-metric of the MPC score[38]. In this study, we observed in Fig. 4c that the MisB class of most severe missense variants contained potential DoS

variants that were overtransmitted in phenotypic probands. In a local, three-dimensional protein structure context, further useful metrics may be derived from identification of areas of evolutionarily constrained sequence that are co-located in the folded protein, but not necessarily adjacent in the primary sequence. The Missense Tolerence Ratio (MTR)[39,40] and Contact Set Missense Tolerance score (COSMIS)[41] are calculated by extension of a sphere of 5 to 8 Angstroms from a given amino acid and quantifying sequence constraint of all the amino acids that project into this local sphere. Interestingly, genes under strong functional constraint show much greater proportions of their encoded protein with constrained regions, calculated by the COSMIS score[41]. This is subtly different to, yet highly compatible with, our finding that functionally constrained proteins are more able to accommodate amino acid substitutions without incurring large structural stability effects.

This work does rely on a single method, MAESTRO, for the exhaustive predictions of mutation stability effects. We replicated a small, representative set of predictions with two other predictors (DynaMut2[26] and DDGun[27]) and the concordance was encouraging, though not exhaustive. The MAESTRO tool has been shown to perform very well and better than Rosetta[42] or FoldX[43] modelling in two comparison studies[11,18]. For exhaustive prediction of stability effects of all possible single amino acid substitutions in the human proteome, one essential consideration was computational expense. For this task, MAESTRO required just 448 CPU days of processor time. Other efficient tools with similar accuracy, such as DynaMut2, require prohibitively more computation time (approximately 7000 CPU years of processor time) to compute a replicate exhaustive prediction set. Though such an exercise to produce this comparative dataset, or improvements in the predictive performance of faster toolsets[18], would facilitate a useful comparison with this data.

All tools used in this study found a narrow range of tolerated stability effects ($-0.5$ kcal/ml $< \Delta\Delta G < 0.5$ kcal/mol) of observed missense variation. Variants with stability effects larger that these thresholds are not generally tolerated and will become subject to purifying selection in functionally constrained genes. We used these thresholds to classify missense variation with outside threshold stability effects as DoS variation. The range of tolerated stability effects was similar between functional constrain measures (pLoF LOEUF and $S_{het}$). The $\Delta\Delta G$ thresholds derived here, that focus on the limits of instability that cause a missense variant to become subject to purifying selection may be very sensitive compared to those that can detect molecular dysfunction from laboratory-based assay. And, indeed, in many contexts the functional perturbation of protein structure may

only need to be very slight for it to become subject to purifying selection. Furthermore, in previous work, we have shown that many missense variants at highly conserved sites had no experimentally assayable phenotype, even when they were in disease genes with an expected mutant phenotype[44].

The stability effects of amino acid substitutions on protein structures are known to be universal among the kingdoms of life[36]. However, the magnitude of the functional effects of missense variation is also appreciated to vary among functional classes of genes[8,9,45], and is potentially related to the severity of disease phenotypes and functional constraint to which different genes can produce and are subject to. One area of future work might investigate whether a diversity of appropriate instability thresholds might be better for different classes of proteins and/or folds than a single threshold value. One example is that it may be better to have differing ΔΔG thresholds for different protein functional contexts, such as the differences between globular proteins and those embedded in membranes. Potentially, too, this will allow annotation of missense variation with more specific predictions of mechanistic labels for individual amino acid substitutions. Further work should seek to refine the thresholds of functional disruption through protein instability, initially by protein functional classifications.

Quantitative estimates of gene functional constraint provide very useful information for interpretation of the importance of coding genetic variation. Here we identify that DoS variation is another reliable source of variation that could be used for calculating an orthogonal metric of functional constraint. The IHS, observed and expected values DoS and predicted ΔΔG scores may be potentially integrated with GeneBayes[14] to iteratively improve estimates of $S_{het}$. Furthermore, the increased resolution provided by missense variant stability effects may mean that that some existing phenotype cohorts are sufficiently large for making population-driven assessments of altered gene constraint in certain phenotypic groups.

## Method
### Datasets
Human population missense variation was obtained fromGnomAD v4.1[12,28]. Canonical transcripts for human genes were obtained from EnsEMBL[46] release GRCh38 p14. Variant calls from disease-free individual genomes were obtained from the 1000 Genomes, Phase 3 release[20]. Annotated pathogenic and benign missense variants were obtained from the ClinVar database[19] release 230930. Predictions of the stability effects of all possible single amino acid substitutions were made using structural models available from the AlphaFold2 Protein Structure Database[17]. The results of MAESTRO predictions[16] made for all human proteins[15] are available at www.stabilitysort.org/download/. 11198 de novo, missense mutations identified among ASD probands were obtained from the original work describing these[31] (see Supplementary Table 20 in Fu et al.[31]). Observed and expected pLoF variant counts and LOEUF scores were obtained from GnomAD[12]. AlphaMissense median pathogenicity scores were obtained from Cheng et al.[21]. $S_{het}$ scores were obtained from the supplementary data of Zeng et al.[14].

### Toolsets
Predictions of variant functional effect were made with PolyPhen2[22], SIFT[23] and CADD[37], obtained using EnsEMBL Variant Effect Predictor[47]. Predictions of the stability effects of single amino acid substitutions on protein structures were performed with MAESTRO[16], DynaMut2[26] and DDGun[27].

### Instability heat scores
To quantitate the per protein instability sensitivity to single amino acid changes, we created the Instability Heat Score (IHS). This score is the average of absolute stability change ($|ΔΔG|$) of all possible amino acid substitutions above the DoS threshold of 0.5:

$$IHS = \frac{1}{S}\sum_{i \in S}|\Delta\Delta Gi| \qquad (1)$$

where $S$ is the set of all possible amino acid substitutions where the absolute stability change is above the specified ΔΔG threshold of 0.5, and $ΔΔGi$ is the stability change associated with the $i$-th possible amino acid substitution.

### LOEUF calculation from DoS variation
To calculate the expected number of DoS variants and the LOEUF score for a given gene, a minor extension was made to existing methodology[12,32]. Briefly, constraint for a given gene is predicted from genomic sequence variation in that gene by quantitating the observed variation in a large population of individuals against estimates of the expected variation, given empirical trinucleotide mutation rates. The existing methodology quantitates observed/expected ratios of unambigious pLoF mutations for a gene. The LOEUF value represents the 90% confidence interval of the ratio of observed and expected pLoF variation for a given gene. In our modification of this approach, substituted pLoF variation with DoS variation, appraising each potential missense substitution for its' predicted effect it on structural stability of a given protein structural model. A substitution is counted as DoS if its' predicted stability effects were greater than defined thresholds (−0.5 <ΔΔG< 0.5, see Results). The estimate of expected DoS variants was then used in place of expected pLoF variants in the calculation of LOUEF for each gene. Likewise, for observed variants, the predicted stability effects of each observed missense variant were appraised and counted if outside of the threshold values being applied. A python implementation of this method using DoS variation is available both as a Zenodo archive[48] and from GitLab (gitlab.com/tdaadt/missensedos).

### Reporting summary
Further information on research design is available in the Nature Portfolio Reporting Summary linked to this article.

## Data availability
The comprehensive set of stability change predictions for all possible single amino acid substitutions in human proteins[15] are available from www.stabilitysort.org/download. Summary data and DoS metrics computed are provided in a Zenodo deposition (doi.org/10.5281/zenodo.14873427) for each human gene with a 1-to-1 mapping between a canonical EnsEMBL transcript (GRCh38 p14 [https://www.ensembl.org/Homo_sapiens/Info/Index?db=core]) and UniProt protein identifier (www.uniprot.org) with a corresponding AlphaFold2 predicted structure (www.alphafold.ebi.ac.uk). Observed and expected pLoF variant counts and LOEUF scores were obtained from GnomAD [gnomad.broadinstitute.org/data]. Variant calls from disease-free individual genomes were obtained from the 1000 Genomes, Phase 3 release [https://www.internationalgenome.org/data-portal/data-collection/phase-3]. Annotated pathogenic and benign missense variants were obtained from the ClinVar database [www.ncbi.nlm.nih.gov/clinvar]. Source data are provided with this paper.

## Code availability
A code implementation for calculating DoS variation and associated metrics is available as a Zenodo archive[48] and a GitLab repository (gitlab.com/tdaadt/missensedos).

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

## Acknowledgements

We thank the National Computational Infrastructure (Australia) for continued access to significant computation resources and technical expertise. AC acknowledges the support of BioPlatforms Australia. This work has been partly funded by a Medical Research Futures Fund (Australia) grant 2016149 (M.M., T.D.A.).

## Author contributions

T.D.A. designed research, T.D.A., M.M., A.C., N.L., L.G., and V.C. performed research, and T.D.A. wrote paper.

## Competing interests

The authors declare no competing interests.
