## [Transparent Peer Review file · Nature Communications]

Functionally constrained human proteins are less prone to mutational instability from single amino acid substitutions

Corresponding Author: Dr Thomas Andrews

Version 0:

Reviewer comments:

Reviewer #1

(Remarks to the Author)

In this study, the authors investigate the stability perturbing effects of missense mutations in the human population. This is an interesting and important topic. However, I am not convinced of the authors' central claim, that the most 'functionally-constrained' human proteins are also the most robust to missense-induced instability. Figure 2A shows an incredibly strong trend: the most constrained human proteins, as defined by gnomAD LOUEF (bin 0) tend to hardly have any 'instability sites' (median ~0.15), while in the least constrained (bin 9), nearly every residue is an instability site (median >0.9). This is remarkable, but in my opinion, it also feels unbelievable - I think it must be confounded by something. Considerable work would need to be done to convince me that this is real.

One aspect I would suggest is essential to understanding this phenomenon is investigating how robust the trend is to different loss-of-function constraint metrics. LOEUF, being the upper bound of a confidence interval of a ratio, can have some funny properties. Is a similar trend seen with pLI? I would also strongly recommend testing it with the recently introduced Shet metric (<https://www.biorxiv.org/content/10.1101/2023.05.19.541520v1>), which seems to have overcome the issues associated with gene size. If a similar trend is also observed for these metrics, it would be more believable.

Second, this is all based on a single stability predictor, MAESTRO. Perhaps the trend is related to MAESTRO's behaviour (maybe it's treatment of disordered regions in AlphaFold structures). It is notable that MAESTRO was previously shown to rank quite poorly against other stability predictors for its ability to distinguish ClinVar pathogenic from gnomAD missense mutations, while forcefield-based methods like FoldX and Rosetta do much better (PMID: 32958805). There is a supplemental figure showing a comparison to Dynamut and DDGun for a small subset of mutations, but to me, those distributions look radically different. I think it is essential to replicate this result with another stability predictor (ideally FoldX or Rosetta). Although this would be computationally expensive, if the trend really is so strong as shown in Fig. 2A, then only a subset of mutations would be needed to show it. In addition, precomputed values from the Rosetta-based RASP (<https://elifesciences.org/articles/82593>) could be used.

As mentioned above, I suspect disordered regions in AlphaFold structures could be an issue. For example, a lot of haploinsufficient genes are transcription factors, which often have large disordered regions. Although it is stated that there is no correspondence with functional constraint and pLDDT, I think this needs to be shown. For example, plot distributions of pLDDT, and fraction of residues with pLDDT >50, vs constraint...

I'm not sure PoPIS is the best metric for robustness. It requires just a single amino acid substitution to be above the threshold to count as an instable site. There is no consideration for the fact that most single amino acid substitutions are not possible by single nucleotide changes, and therefore are unlikely to occur. It could also lead to strong sequence biases, where certain amino acids are far more likely to be classified as instable than others, so it would be helpful to control for amino acid type. I also would expect that this robustness phenomenon should be reflected in the distribution of all mutations, not just the % with a single possible mutation above the threshold.

Although the authors provide python scripts, they should also provide their calculated datasets, included calculated PoPIS values across all proteins, to allow this work to be more easily explored and replicated.

(Remarks on code availability)

Reviewer #2

(Remarks to the Author)

Remarks to the Author:

Summary:

This paper by Andrews and coworkers presents an analysis of the relationship between 1) the effect of missense variants on protein stability, using delta-delta-G predictions, and 2) the degree of functional constraint of each gene. Starting from a previously published dataset of exhaustive ddG predictions of missense variants in AlphaFold models, the authors determine an effective threshold of destabilization of $|\text{ddG}| \geq 0.5$ kcal/mol as calculated by Maestro, for DoF variation (a useful alternative to LoF). They find that proteins with higher functional constraint tend to be more robust to effects of destabilizing missense variants. And they use DoF in place of LoF to better calculate LOEUF scores. The paper is well written, clear, and accessible. It addresses an interesting question and the findings are useful to the field.

Major Comments:

1. The paper relies heavily on LOEUF calculations, which are described briefly in the methods. A clear explanation in the introduction would help the reader follow the argument without needing to look up the original publication.
2. The ddG calculations by Maestro and other methods have been benchmarked against experimental data when run on primarily high-resolution X-Ray crystal structures. Running them on predicted structural models (even good ones by AlphaFold2) introduces an additional level of uncertainty. At the protein level, the confidence can range from very high to very low. At the amino-acid level, many models have regions that are extremely low-confidence. Are the numbers from the ddG calculations weighted in any way to avoid diluting the reliable predictions with the noise? This issue should at least be discussed.

Minor comments:

1. In the first paragraph of Results, “exhaustive stability change data” should be “exhaustive stability change predictions.”
2. The authors state “Proteins from the most functionally constrained categories have significantly higher B-score values.” This statement addresses one form of conformational motion- flexibility. But something should also be said about variants which shift the thermodynamic equilibrium between metastable microstates. This is most important in the case of proteins that were crystallized in more than one conformation. Some of the variants will preferentially stabilize or destabilize one conformation or the other. AlphaFold predictions will represent only one state. How are these cases handled?
3. The discussion of False Discovery Rates is not entirely clear to me. AF2, PolyPhen2, and SIFT are said to give higher values for FDR among ClinVar benign variants. Could that be related to errors in ClinVar, or because those predictors incorporate data other than destabilization/over-stabilization, or some other reason? Does this say anything about variants that affect e.g. protein-protein interaction or post-translational modification?
4. In the results section, HLA-B and HLA-DRB1 are not subunits of the T-cell receptor- rather, they encode MHC Class I and MHC Class II proteins involved in antigen presentation.
5. In Discussion, “more capable of accommodating amino acid substitutions has been observed previously” could benefit from some citations.
6. Some mention could be made of the variants included in GnomAD that are pathogenic but not producing a disease phenotype due to incomplete penetrance, sampling before age of onset, etc.
7. Related to major point #2 above, an analysis of the structural context of each variant could improve the interpretation of each variant. E.g. would it be possible to leverage the structural alphabet used by AF2 to predict a “mechanism” underlying the ddG at each position, or at least rationalize high and low ddG values based on their location within the structure?
8. In figure 2c and 2d, it is difficult to assess the point density due to overlap. If you could provide the ratio of mutants with POPIS scores between 0.9-1 versus 0-0.1, that might help support the statements about “almost all having low POPIS values” and “almost devoid of missense DoF mutations.”
9. In figure 3a and 3b, I don't see a legend explaining the colors of the dots. And adding histograms parallel to the x- and y- axes would make it easier to compare the distributions.

(Remarks on code availability)

Reviewer #3

(Remarks to the Author)

Andrews et al present an analysis of the impact of stability-disrupting missense variants, applying methods that predict stability disruption and estimating depletions of variation for these variants (LOEUF, indicating natural selection). The message is quite straightforward and sensible, that constrained genes are not only depleted of destabilizing variation, but also the available space of single amino acid substitutions is also depleted as such. The idea is certainly very interesting and likely true, but it seems a bit underdeveloped. I have a few comments to improve the clarity of the manuscript and some additional analytic suggestions that could improve the impact as well.

Major comments:

- It would be good to clarify the purpose of this analysis, whether it is simply showing an interesting phenomenon, or proposing a new gene-level pathogenicity score. Figure 3 implies that the latter, but short of the correlation between LOEUF and DoF, there is no indication about whether it outperforms LOEUF (e.g. in identifying haploinsufficient or essential genes). To be clear, I don't think this metric needs to outperform LOEUF to be interesting, since ddG has an interpretation on its own beyond just a pathogenicity score. But it would be good to see where it lies in the distribution.
- Even better, the correlation suggests that similar information is being captured, with DoF more powered at small genes - could the two be combined into a new score that takes both into account?
- The PoPIS metric is interesting, but the jump back and forth between this and "fraction outside -0.5 to 0.5" makes it a little hard to follow. Could these two metrics be shown side in some way to get across the difference: is PoPIS better than the other? I would have thought the empirical observation would have been better.
- Figure 2C: The difference between LOEUF and DoF is a confusing metric. Instead, could you explicitly look at smaller genes and see whether DoF is adding any information? For small genes, is DoF capturing haploinsufficient disease genes?
- I'm glad to see ASD data used as this is an excellent application, but I'm struggling to see what the conclusion is. Is there any enrichment of destabilizing variation among ASD cases?
- I would have liked to see a more thorough comparison to previous literature. In particular, it would be good to compare to the RaSP paper (Blaabjerg et al 2023 eLife), or metrics of missense depletion such as MPC (Samocho et al 2017 bioRxiv).

Minor comments:

- I appreciate the distinguishing of DoF from LoF, but I wonder if stability-disrupting (or DoS) might just be a clearer moniker. There are more ways to affect function than stability.

(Remarks on code availability)

Version 1:

Reviewer comments:

Reviewer #1

(Remarks to the Author)

Overall, I think the authors have done a good job of responding to my original comments, in particular taking into consideration the issue of disorder in AlphaFold structures, and including the Shet constraint metric. I do disagree somewhat that disorder is a "mechanism by which mutation effects are muted" rather than a confounding factor, as I don't think it makes much sense thinking about stabilization/destabilization in a disordered region - given that stability is essentially defined as the difference between a folded and unfolded state. One could probably make most of the conclusions from this paper just by considering disorder, and assuming that mutations in disordered regions are not going to be structurally disruptive. Nevertheless, I do think there are some interesting trends shown in this paper and it is worth publishing, but there is one important issue I think needs to be addressed.

Specifically, the much weaker trend observed for the Shet analysis makes me suspect that the analyses are strongly confounded by length. This is because constraint metrics are known to be strongly confounded by length, and that this is the problem Shet attempts to solve. So, I think the authors should attempt to control for protein length. For ddG, instability and B-factor analyses, how do the trends look if you split into short, medium and long proteins? This could be included as supplemental figures.

(Remarks on code availability)

Reviewer #2

(Remarks to the Author)

(Remarks on code availability)

Reviewer #3

(Remarks to the Author)

The authors have addressed all my comments

(Remarks on code availability)

Version 2:

Reviewer comments:

Reviewer #1

(Remarks to the Author)

I am happy with the authors' response to my previous comments.

(Remarks on code availability)

REVIEWER COMMENTS

Reviewer #1 (Remarks to the Author):

In this study, the authors investigate the stability perturbing effects of missense mutations in the human population. This is an interesting and important topic. However, I am not convinced of the authors' central claim, that the most 'functionally-constrained' human proteins are also the most robust to missense-induced instability. Figure 2A shows an incredibly strong trend: the most constrained human proteins, as defined by gnomAD LOUEF (bin 0) tend to hardly have any 'instability sites' (median ~0.15), while in the least constrained (bin 9), nearly every residue is an instability site (median >0.9). This is remarkable, but in my opinion, it is also feels unbelievable - I think it must be confounded by something. Considerable work would need to be done to convince me that this is real.

Response: We thank this reviewer for their candour and making this important point. We have made major revisions to our analysis to focused on the possible mechanism that produced this trend. We also have softened the language we have used to talk about 'muted effects of mutation in essential proteins' rather than 'robustness to mutation'.

The effect we have observed is at least partially due to the well-accepted feature of essential proteins that they have increased intrinsic disorder. Amino acid substitutions to intrinsically disordered regions have smaller effects of protein stability than in buried regions. But this is more a mechanism by which mutation effects are muted in essential proteins, rather than a confounding factor in our analysis. This remains congruent with our original interpretation, that when seeking causal, pathogenic variation from personal genome data, that the effects of missense mutation in functionally constrained genes (measured by LOEUF or Shet) will be less than in functionally redundant genes, as one of the major mechanisms of pathogenicity (protein instability) is decreased.

We agree with this reviewer that this is a very interesting subject and we have made extensive changes to the manuscript. Here we represent the most obvious change to our Results and Discussion, but have propagated many smaller changes throughout the manuscript. Firstly, to summarise in the Discussion:

Discussion – first para:

“From an exhaustive computational mutagenesis experiment and the predictions of the resulting changes to protein stability, we have observed that, in aggregate, the most functionally constrained human proteins have muted changes to overall protein stability due to the effects of single amino acid substitutions. That some proteins are structurally more capable of accommodating amino acid substitutions has been the subject of long interest (see ³⁴). Both the higher intrinsic disorder that typifies constrained, haploinsufficient genes³⁵ and the higher network centrality of constrained gene^{36,37} go some way to explaining this trend. Smaller proteins with less buried residues are also more robust to mutational instability^{34,38}, and it follows that proteins with a greater proportion of intrinsic disorder will also have relatively fewer buried residues. In addition, it is important to consider the often-lethal effect of mutations to essential, constrained proteins and the subsequent high cost to organismal fitness. These effects may have led to selective pressure driving these proteins towards relatively fewer contacts between residues and therefore increased inherent flexibility to amino acid substitution. This is reflected in the higher median B-factors we observed from experimentally solved structures in the most functionally constrained proteins. It is striking how interwoven this property (captured by IHS) is with the overall functional constraint to which a given gene/protein is subject.”

The major changes in the Results include a new metric to reflect the sensitivity of a protein to instability change:

Results – Section: ‘*Proteins subject to strong functional constraint exhibit muted mutational instability*’ – first sentence:

“We developed an aggregate protein metric, the Instability Heat Score (IHS), to quantitate the susceptibility to mutational stability effects of any given protein, subject to single amino acid substitution mutations.”

With this metric we re-evaluated the trend apparent between mutational instability and functional constraint:

Results – Section ‘*Proteins subject to strong functional constraint exhibit muted to mutational instability*’ – second paragraph:

“We observed that when proteins are partitioned into constraint decile bins by LOEUF that more constrained proteins have broadly lower IHS and therefore are generally less prone to large changes in stability due to missense mutations (Figure 3a). When we included only proteins that have well-powered LOEUF estimates²⁹ (GnomAD v4.1), the differentiation of IHS with functional constraint is slightly decreased (Figure 3b), yet the overall trend towards generally lower IHS in constrained proteins remains. Seeking a potential mechanism for this trend, we considered the contribution of relative protein disorder with IHS. Figure 3c shows a decrease in structural order (increased intrinsic disorder) quantified by the percentage of the AlphaFold2 model for each structure with pLDDT greater than 70%. Visually, this mirrors lower IHS in constrained protein LOEUF deciles groups.”

And we further developed our analysis with pLi and Shet metrics (also see related response below) and B-factors:

Results – Section ‘*Proteins subject to strong functional constraint exhibit muted to mutational instability*’ – last paragraph:

“The broad phenomenon that the most functionally constrained proteins may be less prone to instability by some metrics is recapitulated in the median B-score values of proteins with experimentally solved structures (Figure 3f, Supplementary Data 1). Noteworthy to this investigation is that the experimentally solved structures are frequently only partial structures (see Supplementary Data 1) and do not include intrinsically disordered regions, due to the inherent difficulty with experimental crystallization of these regions. Proteins from the most functionally constrained categories have higher median B-score values than those of the least constrained categories. These proteins in experimentally-solved structures have greater uncertainty (and thus greater movement) of the positions of atoms in the protein crystals. This same trend is also present among gene categories (strong, weak and neutral selection) partitioned by S_{het} score (Figure 3f), though B-factors for proteins in the extreme selection show the broadest variation and encompasses that of all of the other three categories.”

One aspect I would suggest is essential to understanding this phenomenon is investigating how robust the trend is to different loss-of-function constraint metrics. LOEUF, being the upper bound of a confidence interval of a ratio, can have some funny properties. Is a similar trend seen with pLI? I would also strongly recommend testing it with the recently introduced Shet metric (<https://www.biorxiv.org/content/10.1101/2023.05.19.541520v1>), which seems to have overcome the issues associated with gene size. If a similar trend is also observed for these metrics, it would be more believable.

Response: This was an excellent suggestion and we have incorporated these additional metrics into our Results:

“Quantities such as pLi³⁰ provide an alternative metric with which to assess the relationship of IHS with functional constraint³¹. The pLi score observed for all human genes has a strongly bimodal distribution³¹ as shown in the lower distribution in Figure 3d, with a greater number of genes displaying low functional constraint, as expected³¹. We partitioned genes by low (pLi < 0.5) and high (pLi > 0.5) functional constraint and observed the range of IHS for each category (upper boxplot in Figure 3d). Similar to LOEUF decile grouping, the two pLi groups indicate that the higher constraint genes (pLi > 0.5) have a general property of lower IHS, as indicated by the non-overlapping notches between the two boxes. This is despite the two categories containing a very large number of genes each, potential diluting any subtle signals present in this information.

Similarly, we replicated this analysis using the gene selection pressure categories defined by S_{het}¹⁴. Figure 3e shows boxplots of IHS by S_{het} selection constraint categories (extreme, strong, weak and neutral selection) following the naming scheme of Zeng *et al*¹⁴. As with the pLi comparison above, the number of genes present in each selection constraint category varies, and there are very few genes that are neutral. However, the median IHS for genes subject to extreme selection is distinctly lower than the strong and weak categories, though the small number of genes in the neutral category yields an inconclusive comparison with other categories.”

Second, this is all based on a single stability predictor, MAESTRO. Perhaps the trend is related to MAESTRO's behaviour (maybe it's treatment of disordered regions in AlphaFold structures). It is notable that MAESTRO was previously shown to rank quite poorly against other stability predictors for its ability to distinguish ClinVar pathogenic from gnomAD missense mutations, while forcefield-based methods like FoldX and Rosetta do much better (PMID: 32958805). There is a supplemental figure showing a comparison to Dynamut and DDGun for a small subset of mutations, but to me, those distributions look radically different. I think it is essential to replicate this result with another stability predictor (ideally FoldX or Rosetta). Although this would be computationally expensive, if the trend really is so strong as shown in Fig. 2A, then only a subset of mutations would be needed to show it. In addition, precomputed values from the Rosetta-based RASP (<https://elifesciences.org/articles/82593>) could be used.

Response: This is an interesting point, and we have been interested in computing a genome/proteome-wide exhaustive set of predictions. Though this is very computationally expensive, and hence our choice to use MAESTRO. This reviewer's comment is that a representative subset of replicated $\Delta\Delta G$ predictions is useful (and we agree, including small DynaMut2 and DDGun mutation sets), but using FoldX or Rosetta would be essential. We based our choices on a systematic comparison of $\Delta\Delta G$ predictors (<https://doi.org/10.1093/bib/bbac025>), which indicates that both FoldX and Rosetta have lower performance than the tools we have used. Our comparison of predictive methods was motivated to check the narrow range of DDG values we observed from the MAESTRO tool. We have moved the comparison of DynaMut2, DDGun and MAESTRO from the supplemental figures and changed the Results section to better state that the three tools also showed a narrow range of instability values (compared to empirical values):

Results- Section “*Instability thresholds for missense disruption-of-function variants*” – paragraph 3:
“These $\Delta\Delta G$ threshold values ($-0.5 < \Delta\Delta G < 0.5$) are narrow compared to those empirically measured^{25,26}. For comparison, we replicated $\Delta\Delta G$ predictions for 2419 amino acid substitutions observed across eleven proteins with

two additional tools: DynaMut2²⁷ and DDGun²⁸. Both additional tools have been shown to provide similar accuracy to MAESTRO¹¹. The predicted values of these additional tools also show the narrower range of predicted $\Delta\Delta G$ values than that observed in empirical data (Figure 1e), lending support to the finding above that at relatively minor predicted instability values, there are observable depletions of missense variants among functionally constrained genes compared to those that are relatively unconstrained.“

As suggested by this reviewer, the manuscript for the RaSP tool (<https://elifesciences.org/articles/82593>) also includes an independent comparison of predictors, with similar results as the systematic comparison study listed above.

In the training data for the RaSP manuscript, we explored the Rosetta predictions of $\Delta\Delta G$. These, however, are mostly not from human proteins or even mammalian species, and whilst the stability effects of mutation on proteins seem universal, they are somewhat tangential to our use-case, and we have relied on the results of the systematic tool comparisons from the literature. As suggested, something of strong interest was the genome/proteome-wide exhaustive prediction set that was generated with the RaSP tool. After extensive re-analysis, we did not include these results, for the following reason. The RaSP $\Delta\Delta G$ predictions did not resolve the constraint-based selection against high-instability amino acid substitutions, as shown in Figure 1A&B with both LOEUF & Shet constraint categories. One orienting assumption is that the range of observed $\Delta\Delta G$ values for a given protein should reflect the degree of functional constraint to which that protein is subject. This should not be constant across proteins of different constraint. And is a fundamental property of any $\Delta\Delta G$ predictor that it should be able to resolve a strong signal such as this. As the focus of this manuscript was not intended to be an systematic appraisal of existing methodology (though would be warranted in this fast-moving area, including the new progress made with Shet scores), we excluded our analysis with RaSP $\Delta\Delta G$ predictions. Though we remain attentive further RaSP versions or similar tool release to repeat this appraisal.

As mentioned above, I suspect disordered regions in AlphaFold structures could be an issue. For example, a lot of haploinsufficient genes are transcription factors, which often have large disordered regions. Although it is stated that there is no correspondence with functional constraint and pLDDT, I think this needs to be shown. For example, plot distributions of pLDDT, and fraction of residues with pLDDT >50, vs constraint...

Response: We agree, and much of our re-analysis due to this point is also included in our response to this reviewers first point above. And we thank this reviewer for refining our investigation to use a metric such as the pLDDT>50 for ordered/disordered classification of protein regions. We did just as suggested (though used pLDDT>70) in our reanalysis. This is shown in the revised Figure 3c, which does now indicate a distinct correlation of protein order with functional constraint. This refinement has driven much of the major revision of this manuscript. We not conclude that variation of intrinsic disorder with functional constraint is at least partially the mechanism behind the muted effects of missense mutation in highly constrained genes.

I'm not sure PoPIS is the best metric for robustness. It requires just a single amino acid substitution to be above the threshold to count as an instable site. There is no consideration for the fact that most single amino acid substitutions are not possible by single nucleotide changes, and therefore are unlikely to occur. It could also lead to strong sequence biases, where certain amino acids are far more likely to be classified as instable than others, so it would be helpful to control for amino acid type. I also would expect that this robustness phenomenon should be reflected in the distribution of all mutations, not just the % with a single possible mutation above the threshold.

Response: We have removed the PoPIS metric from the revised manuscript and replaced it with the Instability Heat Score. The new metric avoids the yes/no treatment of possible instability at protein sites, as pointed out, but now takes into account the magnitude of predicted stability for each possible amino acid substitution. We have not made a distinction between amino acid substitutions that require a single or more than a single nucleotide change (though this is an interesting suggestion) for the reasons that: i) this requires definition of a reference sequence for each gene, which makes assumptions about the stability of a given reference protein (and the reference human genome is a small sample of genomic diversity), which is beyond the scope of this present work. Though it definitely has not escaped our interest that a survey of stability variation across thousands of genomes would be very useful for defining the disease-free individuals, and ii) as the DDG predictions have an inherent uncertainty, this variation is almost certainly greater than the refinement of single/multiple nucleotide changes baked into the IHS. A very interesting comment and definitely a topic for future work.

Although the authors provide python scripts, they should also provide their calculated datasets, included calculated PoPIS values across all proteins, to allow this work to be more easily explored and replicated.

Response: the supplementary data (Supplementary Data 2) for the revised manuscript now includes the ISH and other metrics.

Reviewer #2 (Remarks to the Author):

Remarks to the Author:

Summary:

This paper by Andrews and coworkers presents an analysis of the relationship between 1) the effect of missense variants on protein stability, using delta-delta-G predictions, and 2) the degree of functional constraint of each gene. Starting from a previously published dataset of exhaustive ddG predictions of missense variants in AlphaFold models, the authors determine an effective threshold of destabilization of $|ddG| \geq 0.5$ kcal/mol as calculated by Maestro, for DoF variation (a useful alternative to LoF). They find that proteins with higher functional constraint tend to be more robust to effects of destabilizing missense variants. And they use DoF in place of

LoF to better calculate LOEUF scores. The paper is well written, clear, and accessible. It addresses an interesting question and the findings are useful to the field.

Major Comments:

1. The paper relies heavily on LOEUF calculations, which are described briefly in the methods. A clear explanation in the introduction would help the reader follow the argument without needing to look up the original publication.

Response: We have included an expanded description of LOEUF calculations in the Introduction, whilst needing to remain concise for space reasons. We also include an expanded description of LOEUF calculations and our extensions to this in the Materials and Methods.

Introduction – paragraph 4:

To quantitate functional constraint, the ‘loss-of-function observed/expected upper bound fraction’ (LOEUF) score and the relative rank of LOEUF score among all genes¹² (with genes commonly partitioned bins by decile) has been produced with the GnomAD dataset¹². Briefly, the LOEUF value represents the 90% confidence interval of the ratio of observed and expected pLoF variation for a given gene, where the expected pLoF variation is calculated from empirical trinucleotide mutation frequencies and the composition of these in each given gene¹². The LOEUF scores calculated from population-scale genome datasets count only the most unambiguous of potential pLoF mutants: nonsense, frameshift and canonical splice donor-acceptor mutants¹³.

Materials and Methods – Section ‘*LOEUF Calculation from DoS variation*’

To calculate the expected number of DoS variants and the LOEUF score for a given gene, a minor extension was made to existing methodology^{12,33}. This methodology was built to estimate the expected numbers of pLoF mutations for a gene, given a model of the relative trinucleotide mutation frequencies in non-coding regions, calibrated with the numbers of observed synonymous substitutions for that gene. From this, and for potential pLoF sites for a given sequence, the expected number of substitutions for that type may be estimated. In our modification of this approach, we appraised each potential missense substitution for its’ predicted effect it on structural stability of a given protein structural model. A substitution is counted as DoS if its’ predicted stability effects were greater than defined thresholds ($-0.5 < \text{DDG} < 0.5$, see Results). The estimate of expected DoS variants was then used in place of expected pLoF variants in the calculation of LOEUF for each gene.

2. The ddG calculations by Maestro and other methods have been benchmarked against experimental data when run on primarily high-resolution X-Ray crystal structures. Running them on predicted structural models (even good ones by AlphaFold2) introduces an additional level of uncertainty. At the protein level, the confidence can range from very high to very low. At the amino-acid level, many models have regions that are extremely low-confidence. Are the numbers from the ddG calculations weighted in any way to avoid diluting the reliable predictions with the noise? This issue should at least be discussed.

Response: this is an insightful comment and is closely related to the mechanism that we propose is related to the muted stability effects of amino acid substitution we have characterised in functionally constrained proteins. The revised manuscript has incorporated a new metric of the order observed in the AlphaFold2 models, derived from the pLDDT scores that can be used as model confidence scores. One remarkable feature of the AlphaFold2 models is that they create a quasi map of regions of intrinsic disorder for each protein model. We have included analysis of the in the revised Results and expand on this in the revised Discussion.

Results – Section ‘*Proteins subject to strong functional constraint exhibit muted mutational instability*’ – paragraph 2:

We observed that when proteins are partitioned into constraint decile bins by LOEUF that more constrained proteins have broadly lower IHS and therefore are generally less prone to large changes in stability due to missense mutations (Figure 3a). When we included only proteins that have well-powered LOEUF estimates²⁹ (GnomAD v4.1), the differentiation of IHS with functional constraint is slightly decreased (Figure 3b), yet the overall trend towards generally lower IHS in constrained proteins remains. Seeking a potential mechanism for this trend, we considered the contribution of relative protein disorder with IHS. Figure 3c shows a decrease in structural order (increased intrinsic disorder) quantified by the percentage of the AlphaFold2 model for each structure with pLDDT greater than 70%. Visually, this mirrors lower IHS in constrained protein LOEUF deciles groups.

Results – Figure 3c:

Plot shows: **c)** proportion of protein (AlphaFold2-model) with order greater than pLDDT > 0.7 by LOEUF functional constraint decile bin,

Discussion – paragraph 1:

From an exhaustive computational mutagenesis experiment and the predictions of the resulting changes to protein stability, we have observed that in aggregate the most functionally constrained human proteins have muted changes to overall protein stability due to the effects of single amino acid substitutions. That some proteins are structurally more capable of accommodating amino acid substitutions has been the subject of long interest (see³⁴). Both the higher intrinsic disorder that typifies constrained, haploinsufficient genes³⁵ and the higher network centrality of constrained gene^{36,37} go some way to explaining this trend. Smaller proteins with less buried residues are also more robust to mutational instability^{34,38}, and it follows that proteins with a greater proportion of intrinsic disorder will also have relatively fewer buried residues per unit of protein length. In addition, it is important to consider the often-lethal effect of mutations to essential, constrained proteins and the subsequent high cost to organismal fitness. These effects may have led to selective pressure driving these proteins towards relatively fewer contacts between residues and therefore increased inherent flexibility to amino acid substitution. This is reflected in the higher median B-factors we observed from experimentally solved structures in the most functionally constrained proteins. It is striking how interwoven this property (captured by IHS) is with the overall functional constraint to which a given gene/protein is subject.

Of interest, we also appraise B-factors from high-resolution X-ray crystal structures as an orthogonal dataset. We found a similar trend, though in the absence of intrinsic disorder. We include the following in the Results section:

Results – section ‘*Proteins subject to strong functional constraint exhibit muted mutational instability*’ – paragraph 5:

The broad phenomenon that the most functionally constrained proteins may be less prone to instability by some metrics is recapitulated in the median B-score values of proteins with experimentally solved structures (Figure 3f, Supplementary Data 1). Noteworthy to this investigation is that the experimentally solved structures are predominantly partial structures (see Supplementary Data 1) and do not in general include intrinsically disordered regions. Proteins from the most functionally constrained categories have higher median B-score values than those of the least constrained categories. These proteins in solved structures have greater uncertainty (and thus greater movement) of the positions of atoms in the protein crystals. This same trend is also present among gene categories (strong, weak and neutral selection) partitioned by S_{het} score (Figure 3f), though B-factors for proteins in the extreme selection show the broadest variation and encompasses that of all of the other three categories.

Minor comments:

1. In the first paragraph of Results, “exhaustive stability change data” should be “exhaustive

stability change predictions.”

Response: Yes, this is better and correct. This change has been made.

2. The authors state “Proteins from the most functionally constrained categories have significantly higher B-score values.” This statement addresses one form of conformational motion- flexibility. But something should also be said about variants which shift the thermodynamic equilibrium between metastable microstates. This is most important in the case of proteins that were crystallized in more than one conformation. Some of the variants will preferentially stabilize or destabilize one conformation or the other. AlphaFold predictions will represent only one state. How are these cases handled?

Response: this is an interesting point given that, especially, gain-of-function mutants may cluster at these sites. Though comparatively, these GoF mutants are rare when compared to LoF mutants. This investigation is outside of the scope of the present study, primarily due to the structural data we have relied on being models generated with AlphaFold2. Structural bioinformatics is discouraged of using AlphaFold2 for modelling mutant structures (eg. <https://doi.org/10.1038/s41594-022-00849-w>), and with just the wild-type AlphaFold2 models we have a blind-spot for these changes between metastable microstates. For the time being, we have not addressed the mutational impact in such scenarios, but it would be interesting to explore if proteome-wide data for this became available.

3. The discussion of False Discovery Rates is not entirely clear to me. AF2, PolyPhen2, and SIFT are said to give higher values for FDR among ClinVar benign variants. Could that be related to errors in ClinVar, or because those predictors incorporate data other than destabilization/over-stabilization, or some other reason? Does this say anything about variants that affect e.g. protein-protein interaction or post-translational modification?

Response: In this analysis we have assumed that the ClinVar predictions are a ‘gold standard’ resource and have evaluated these tools in this context. We estimated the FDR on ClinVar Benign variants by comparing accuracy of each tool ($\Delta\Delta G$, AlphaMissense, pph2 and SIFT) in recapitulating this label. For two of these tools (pph2, SIFT), the predictions are primarily driven by observations of inter-species sequence conservation – and are known for their high FDR, as very highly conserved sites are very often not associated with a mutant phenotype when substituted (see: <https://www.pnas.org/doi/10.1073/pnas.1511585112>). It is interesting that AlphaMissense has only a narrow edge over predicted $\Delta\Delta G$, as AM is built to make predictions of mutant effect, rather than stability change. Though, one interpretation of this result is that stability effects are very closely aligned with the mutation effect. PPIs and PTMs are an interesting idea that we intend to follow-up in future work.

4. In the results section, HLA-B and HLA-DRB1 are not subunits of the T-cell receptor- rather, they encode MHC Class I and MHC Class II proteins involved in antigen presentation.

Response: We are grateful to this reviewer for pointing out this error. This now reads:

Results – Section ‘*Comparison of constraint between pLoF and DoS variation*’ – paragraph 4 -

Of note, HLA-B and HLA-DRB1 genes have LOUEF scores calculated with DoS variants that are more than double that calculated with pLoF variation. These genes encode MHC Class I and MHC Class II proteins and their high polymorphism is related to their immune function. This functional polymorphism and does not include LoF polymorphism.

5. In Discussion, “more capable of accommodating amino acid substitutions has been observed previously” could benefit from some citations.

Response: we have now included citations, especially the excellent review of Tokuriki & Tawfik “Stability effects of mutations and protein evolvability”. No recent review has a better discussion of this area. The changes made are:

Discussion – paragraph 1:

“That some proteins are structurally more capable of accommodating amino acid substitutions has been the subject of long interest (see ³⁴). Both the higher intrinsic disorder that typifies constrained, haploinsufficient genes³⁵ and the higher network centrality of constrained gene^{36,37} go some way to explaining this trend. Smaller proteins with less buried residues are also more robust to mutational instability^{34,38}, and it follows that proteins with a greater proportion of intrinsic disorder will also have relatively fewer buried residues per unit of protein length.”

6. Some mention could be made of the variants included in GnomAD that are pathogenic but not producing a disease phenotype due to incomplete penetrance, sampling before age of onset, etc.

Response: Yes, we are at the mercy of the data that is available. This is a primary reason that depletion of genetic variation between LOEUF bin 0 and bin 9 genes was used for identifying thresholds of DDG variation in nominally disease-free individuals. Even though this data will be many variants of the types this reviewer mentions. We have included the following sentence to this effect:

Results – Section: ‘*Instability thresholds for missense disruption-of-function variants*’ -

Our presumption was that among these constrained proteins, variants with large stability effects that affect protein function will become subject to purifying selection and hence be absent from the sequence record of disease-free human populations. And among proteins subject to less functional constraint, variants of similar stability effects will be variably present due to lower purifying selection active in the encoding genes.

7. Related to major point #2 above, an analysis of the structural context of each variant could improve the interpretation of each variant. E.g. would it be possible to leverage the structural alphabet used by AF2 to predict a “mechanism” underlying the ddG at each position, or at least rationalize high and low ddG values based on their location within the structure?

Response: Yes, this is an excellent idea and is related to what we shall work on next: classification of predicted DDG in the context of protein fold classification, allowing focussed thresholds for DoS variation. We include a paragraph about this in the Discussion section containing a comment about mechanistic labels for DDG variation:

Discussion – paragraph 7:

“The stability effects of amino acid substitutions on protein structures are known to be universal among the kingdoms of life³⁸. However, the magnitude of the functional effects of missense variation is also appreciated to vary among functional classes of genes^{8,9,47}, and is potentially related to the severity of disease phenotypes and functional constraint to which different genes can produce and are subject to. One area of future work might investigate whether a diversity of appropriate instability thresholds might be better for different classes of proteins and/or folds than a single threshold value. One example is that it may be better to have differing DDG thresholds for different protein functional contexts, such as the differences between globular proteins and those embedded in membranes. Potentially, too, this will allow annotation of missense variation with more specific predictions of mechanistic labels for individual amino acid substitutions. Further work should seek to refine the thresholds of functional disruption through protein instability, initially by protein functional classifications.”

8. In figure 2c and 2d, it is difficult to assess the point density due to overlap. If you could provide the ratio of mutants with POPIS scores between 0.9-1 versus 0-0.1, that might help support the statements about “almost all having low POPIS values” and “almost devoid of missense DoF mutations.”

Response: We thank this reviewer for this comment, and it has helped us clarify the revised manuscript. We have revised Figure 2c and 2d, which have been moved to a separate, larger Figure 4. We also have replaced the PoPIS scores with the Instability Heat Score. Figure 4a has lesser overlapping points compared to the original Figure 2c. The figure now better illustrates the points made in the text, though do accept that over-plotting in the densest regions of Figure 4b is still present. As suggested by this reviewer, we have now included summary statistics in the text to quantify the variation in the distributions of IHS with mutational instability. This revised text now reads as:

Results – Section “Instability Heat Scores and relative pathogenic effect of missense variation” –

“The 72 ASD-linked genes are almost all from the most functionally constrained classes (54 of 72 genes are from LOEUF decile bin 0 - meaning the top 10% of most constrained genes). The 72 ASD-linked genes are slightly skewed towards lower IHS (65.5% of ASD-linked genes have an IHS of less than 0.6, compared to 51.0% among all genes. Only three of the 72 ASD-linked genes (DEAF1, PTEN, AP2S1) harbor potential DoS variation in ASD probands, yet this variation represents 83.0% of all de novo missense variation in these 72 ASD-linked genes in proband individuals, compared to 15.5% for the set of all genes (Figures 4a&b, Supplementary Table 1). Potential DoS variation in only two ASD-linked genes (PTEN and AP2S1) contributes most to this differential between gene sets. These gene are outliers compared to the other ASD-linked genes, having the highest two IHS of the 72 ASD-associated genes, and are two of the least functionally constrained proteins in this set (Figure 4c). These metrics (BF, IHS and DDG) used together seem a useful means of further segregating genes that may contribute to phenotype through missense variation.”

9. In figure 3a and 3b, I don't see a legend explaining the colors of the dots. And adding histograms parallel to the x- and y-axes would make it easier to compare the distributions.

Response: This figure has moved to an altered Figure 5a in the revised manuscript. We have replotted this figure with the revised dataset and now include the suggested features (including histograms and a colour legend) to the new plot, and the associated legend text.

Reviewer #3 (Remarks to the Author):

Andrews et al present an analysis of the impact of stability-disrupting missense variants, applying methods that predict stability disruption and estimating depletions of variation for these variants (LOEUF, indicating natural selection). The message is quite straightforward and sensible, that constrained genes are not only depleted of destabilizing variation, but also the available space of single amino acid substitutions is also depleted as such. The idea is certainly very interesting and likely true, but it seems a bit underdeveloped. I have a few comments to improve the clarity of the manuscript and some additional analytic suggestions that could improve the impact as well.

Major comments:

- It would be good to clarify the purpose of this analysis, whether it is simply showing an interesting phenomenon, or proposing a new gene-level pathogenicity score. Figure 3 implies that the latter, but short of the correlation between LOEUF and DoF, there is no indication about whether it outperforms LOEUF (e.g. in identifying haploinsufficient or essential genes). To be clear, I don't think this metric needs to outperform LOEUF to be interesting, since ddG has an interpretation on its own beyond just a pathogenicity score. But it would be good to see where it lies in the distribution.

Response: We thank this reviewer for flagging this ambiguity and have endeavoured in the revisions to make this clearer. Our intention was to identify and characterise the relationship between functional constraint (determined by any of LOEUF, pLi or Shet) and the susceptibility of a given protein to mutational instability (in this revision, represented by the Instability Heat Score).

We have included the following revisions that work to explain the main message of this manuscript:

Abstract – “At a proteome-wide scale, we have quantitated potential disruption of protein stability due to amino acid substitution and show that the most functionally constrained proteins are typically less susceptible to large mutational changes in stability. Mechanistically, this relates to greater intrinsic disorder among constrained proteins, but also to increased B-factors in the ordered regions of constrained proteins. This phenomenon means that constrained proteins exhibit smaller stability effects due missense mutations, and partly explains why overtransmission of pathogenic missense variation is less prevalent in genetic disorders characterised with protein truncating variation.”

Introduction – last paragraph: “Here we investigate the relative depletion of missense variants with high predicted protein stability impacts from genes under strong functional constraint. With this, we identify thresholds at which missense mutations disrupt protein stability and become DoS variants. From these we identify that the most functionally constrained genes in the genome are comparatively less prone to mutational instability than functionally redundant proteins.”

Discussion – first paragraph: “From an exhaustive computational mutagenesis experiment and the predictions of the resulting changes to protein stability, we have observed that, in aggregate, the most functionally constrained human proteins have muted changes to overall protein stability due to the effects of single amino acid substitutions.”

Discussion – first paragraph: “One important consequence of this will be that fewer missense variants in strongly constrained disease genes will result in pathogenic effects, due to a central pathogenic mechanism of mutation being muted in these genes. Hence, when searching affected cohorts and/or

personal genomes for causal mutations, if the phenotype or condition results from alteration or ablation of the function of highly constrained genes, the contribution of missense variation may be slight, and it is important to prioritise other mutation types in these investigations.“

The manuscript does not focus on producing a predictor of variant effect, based on mutational instability, we have added a comment in the Discussion that may better point the reader to other resources for this:

Discussion – second paragraph: “We have previously produced a tool, StabilitySort (www.stabilitysort.org), to prioritise potentially pathogenic missense mutation using the the population variation of protein stability, as $\Delta\Delta G$ when combined with a Z score to represent the ‘unusualness’ of the stability effects of a given variant in a given protein is a remarkably useful predictor of pathogenicity¹⁵. ”

- Even better, the correlation suggests that similar information is being captured, with DoF more powered at small genes - could the two be combined into a new score that takes both into account?

Response: This is a great idea. One of the original motivations was to use DoS variation as a better powered alternative to pLoF LOEUF, and this forms the last section in the Results of this original manuscript. The now published framework of GeneBayes, used to calculate Shet estimates that are less underpowered than LOEUF for shorter genes (<https://doi.org/10.1038/s41588-024-01820-9>), represents an excellent means to include $\Delta\Delta G$, IHS and DoS LOEUF metrics to iteratively improve Shet calculations. We have included a comment in the final paragraph of the Discussion section to this effect:

Discussion – final paragraph: “Here we identify that DoS variation is another reliable source of variation that could be used for calculating an orthogonal metric of functional constraint. The IHS, the DoS observed and expected values and the predicted DDG scores may be potentially integrated with GeneBayes¹⁴ to iteratively improve estimates of S_{het} .”

- The PoPIS metric is interesting, but the jump back and forth between this and “fraction outside -0.5 to 0.5” makes it a little hard to follow. Could these two metrics be shown side in some way to get across the difference: is PoPIS better than the other? I would have thought the empirical observation would have been better.

Response: We have improved on the PoPIS metric in the revised manuscript with the Instability Heat Score (IHS), yet this comment is still relevant with this new score also. The $\Delta\Delta G$ thresholds of -0.5 and 0.5 are used in the analysis for classifying stability effects into DoS and benign. Whereas the PoPIS and IHS metrics are values related to the ease with which mutations may cause instability within a specific protein. This distinction was not very apparent in the text of the original manuscript, and we have revised this to make it clearer:

Results – Section “*Instability thresholds for missense disruption-of-function variants* “: “Using this observed depletion of protein instability variation in the most constrained (bin 0) genes, we chose upper and lower $\Delta\Delta G$ cut-offs of 0.5 and -0.5 kcal/mol, respectively. These thresholds represent the instability boundaries beyond which a missense variant will become subject to purifying selection in a functionally constrained gene”

Results – Section “*Proteins subject to strong functional constraint exhibit muted to mutational instability*”:

“We developed an aggregate protein metric, the Instability Heat Score (IHS), to quantitate the susceptibility to mutational stability effects of any given protein, subject to single amino acid substitution mutations. For any given protein, the IHS is the average of the absolute stability change ($|\Delta\Delta G|$) of all possible substitutions that result in a predicted stability change that lies outside the thresholds determined above ($-0.5 < \Delta\Delta G < 0.5$). This metric is built on the thresholds determined in the previous section, yet represents the susceptibility (or fragility) of a protein to instability introduced by mutations that substitution single amino acids. Proteins with higher IHS are fragile to mutational instability, whereas those with low IHS are ‘cold’ an, in general, do not exhibit large mutational instability.”

- Figure 2C: The difference between LOEUF and DoF is a confusing metric. Instead, could you explicitly look at smaller genes and see whether DoF is adding any information? For small genes, is DoF capturing haploinsufficient disease genes?

Response: We accept this reviewers’ point, it was confusing, and have removed this figure from the revised manuscript. In Table 1, we do an analysis in microcosm that is similar to that suggested. We identify example genes (that are in the shortest 25% of genes) that appear unconstrained with pLoF LOUEF, and have few observed pLoF variants, yet by DoS LOEUF have a larger number of observed DoS variants and go from appearing highly constrained to relatively unconstrained.

- I’m glad to see ASD data used as this is an excellent application, but I’m struggling to see what the conclusion is. Is there any enrichment of destabilizing variation among ASD cases?

Response: The mention of the analysis of the ASD genomes was brief in the original manuscript, but we have expanded this in the revised manuscript. To summarise, this analysis is a case study to demonstrate the utility of IHS and how it may indicate that certain proteins are less prone to mutational instability. And it is a clearer result that we expected – and demonstrates the utility of our approach. This analysis has been expanded to a full Results section in the revised manuscript and includes an expanded figure (Figure 4).

The purpose of the inclusion of analysis of this very large genome dataset was motivated by the knowledge that the ASD-linked genes from the original work of Fu et al (2023; <https://www.nature.com/articles/s41588-022-01104-0>) were almost all highly functionally constrained (LOEUF decile categories 0 and 1). The work of Fu et al finds the strongest evidence for overtransmission of loss-of-function variation from protein truncating variation (pLoF), and comparative less from missense variants. We were curious to know the IHS of the ASD-linked proteins and the instability effects of these observed missense variants in the ASD probands. Both were illuminating – i) the ASD-linked proteins almost universally had very low IHS, even among the most constrained proteins, and ii) the observed missense mutants in the ASD probands were within the DoS thresholds. There were two exceptions, in the two genes (PTEN and AP2S1).

The expanded results section now reads as:

Results – Section: “*Instability Heat Scores and relative pathogenic effect of missense variation*”

“We explored the relevance of protein IHS in the context of missense variation observed in a large phenotype-associated genome dataset, obtained from individuals with autism and their families³². This dataset aggregates genomic variation from 63237 individuals and calculates Bayes Factor (BF) values to quantify evidence of

phenotypic association from multiple variant classes (protein truncating variation (PTV), missense variation, indels and copy number variation) and modes of inheritance³². We used these BF values to investigate covariation with both IHS and predicted $\Delta\Delta G$ scores, at both a per gene and per variant level. The BF dataset strongly indicates that PTV contributes most evidence that implicates the majority of the 72 ASD-linked genes³². But intriguingly, for a minority of genes, missense variation provides more than 90% of the evidence that links them to an autism phenotype in proband individuals (eg. PTEN, SLC6A1, DYNC1H1, DEAF1, AP2S1)³². The 72 ASD-linked genes are overwhelmingly from the most functionally constrained classes (54 of 72 genes are from the top 10% of most constrained gene class). We find (Figure 4a) that these 72 genes are heavily skewed towards lower IHS and as a group have some of the lowest IHS values in the genome. Furthermore, only three of the 72 ASD-linked genes (DEAF1, PTEN, AP2S1) harbor potential DoS variation in ASD probands (Figure 4a, Supplementary Table 1). Whereas Figure 4b shows this same comparison for all *de novo* variation identified in all genes (both ASD-linked variation and passenger mutations). Comparison between Figure 4a and 4b identifies that, as a group, ASD-associated genes have i) much lower IHS, ii) much narrower range of predicted DDG, and iii) comparatively almost no *de novo* variants in ASD-linked genes harbor functionally-important, DoS variation. Two ASD-linked genes (PTEN and AP2S1) are outliers to the above generalizations, having the highest two IHS of the 72 ASD-associated genes, and are two of the least functionally constrained proteins in this set. Supplementary Table 1 provides further information of IHS, LOEUF functional constraint and *de novo* missense mutations observed outside DoS thresholds in ASD probands. Figure 4c shows BF values for PTV and missense variation (MisB and MisA classes, after nomenclature of Fu *et al*³²) plotted against IHS. This comparison shows that PTEN and AP2S1 have the highest susceptibility to DoS mutations of the 72 ASD-linked genes, and this is reflected in the unusually high contribution of missense variation (BF values from MisB) to detecting this phenotypic association.”

- I would have liked to see a more thorough comparison to previous literature. In particular, it would be good to compare to the RaSP paper (Blaabjerg et al 2023 eLife), or metrics of missense depletion such as MPC (Samocha et al 2017 bioRxiv).

Response: These are good suggestions. We have rectified the omission of the MPC score and metrics of missense depletion with this addition to the Discussion:

Discussion - paragraph 4:

“Genes and gene regions are variably tolerant of missense variation, and much attention has been directed towards the predictive usefulness of relative depletion of missense variation to identify functionally constrained elements. In linear coding sequences, the regional depletion of missense variation combined with an amino acid substitution score and PolyPhen2 scores has provided the joint-metric of the MPC score⁴⁰. In this study, we observed in Figure 4c that the MisB class of most severe missense variants contained potential DoS variants that were overtransmitted in phenotypic probands”

The omission of mention of the RaSP paper was also pointed out by reviewer 1. Our response to this suggestion was: As suggested, something of strong interest was the genome/proteome-wide exhaustive prediction set that was generated with the RaSP tool. After extensive re-analysis, we did not include these results, for the following reason. The RaSP $\Delta\Delta G$ predictions did not resolve the constraint-based selection against high-instability amino acid substitutions, as shown in Figure 1A&B with both LOEUF & Shet constraint categories. One orienting assumption is that the range of observed $\Delta\Delta G$ values for a given protein should reflect the degree of functional constraint to which that protein is subject. This should not be constant across proteins of different constraint. And is a fundamental property of any $\Delta\Delta G$ predictor that it should be able to resolve a strong signal such as this. As the focus of this manuscript was not intended to be an appraisal of existing methodology (though would be warranted in this fast-moving area, including the new progress made with Shet scores), we excluded our analysis with RaSP $\Delta\Delta G$

predictions. Though we remain attentive further RaSP versions or similar tool release to repeat this appraisal.

Minor comments:

- I appreciate the distinguishing of DoF from LoF, but I wonder if stability-disrupting (or DoS) might just be a clearer moniker. There are more ways to affect function than stability.

Response: We agree and think this is an excellent name for this variation. We have adopted this throughout the manuscript now, as it avoids the comparatively ambiguous DoF term.

Reviewer #1 (Remarks to the Author):

Overall, I think the authors have done a good job of responding to my original comments, in particular taking into consideration the issue of disorder in AlphaFold structures, and including the Shet constraint metric. I do disagree somewhat that disorder is a "mechanism by which mutation effects are muted" rather than a confounding factor, as I don't think it makes much sense thinking about stabilization/destabilization in a disordered region - given that stability is essentially defined as the difference between a folded and unfolded state. One could probably make most of the conclusions from this paper just by considering disorder, and assuming that mutations in disordered regions are not going to be structurally disruptive. Nevertheless, I do think there are some interesting trends shown in this paper and it is worth publishing, but there is one important issue I think needs to be addressed.

Specifically, the much weaker trend observed for the Shet analysis makes me suspect that the analyses are strongly confounded by length. This is because constraint metrics are known to be strongly confounded by length, and that this is the problem Shet attempts to solve. So, I think the authors should attempt to control for protein length. For $\Delta\Delta G$, instability and B-factor analyses, how do the trends look if you split into short, medium and long proteins? This could be included as supplemental figures.

Response:

Once again, we thank this reviewer for their considered opinion and their constructive suggestions to this manuscript. And we do understand and agree with the point made here about disorder. It is interesting that this also correlates with functional constraint.

Addressing the point about protein length, we have included a new panel of supplementary figures that allows the metrics suggested to be appraised by length. As with disorder metrics, length does correlate with functional constraint metrics, especially for LOEUF. As with disorder, this may be argued to be a confounder, but this is also a correlate with increased robustness to mutational instability.

The following changes to the manuscript text have been made:

Results, Pg 5, last para - added:

Supplementary Figures 1a and 1b show that both LOEUF and S_{het} functional constraint categories, respectively, are heterogeneous with respect to gene lengths, with longer genes being associated with higher constraint. Supplementary Figure 1c shows that tolerated levels of mutational instability vary somewhat with protein length, especially among the shortest proteins with fewer than 400 amino acids.

Results, Pg 7, last para – added:

Supplementary Figure 1d shows this same trend in a similar boxplot of IHS calculated using variable instability thresholds, dependent on protein length, which partially normalises for the protein length variation observed between constraint categories (Supplementary Figure 1a).

Results, Pg 9, second para – added:

Supplementary Figure 1d shows this same trend in a similar boxplot of IHS calculated using variable instability thresholds, dependent on protein length, which partially normalises for the protein length variation observed between constraint categories (Supplementary Figure 1a).

Discussion, Pg 12, first para – added and edited:

The size distribution of proteins versus their functional constraint has also received recent attention, and we observe here that longer proteins generally have smaller stability effects due to mutation than shorter proteins. This is despite our understanding that proteins with fewer buried residues are also more robust to mutational instability^{33,37}. It is important to consider this in the context that the often-lethal effect of mutations to essential, constrained proteins have a subsequent high cost to organismal fitness.

Supplementary Figure 1 and figure legend – added:

Supplementary Figure 1 – Relationship between gene/protein length and predicted $\Delta\Delta G$. Density plots of gene length versus functional constraint categories for **a)** GnomAD LOEUF constraint deciles, and **b)** S_{net} constraint categories, **c)** Predicted $\Delta\Delta G$ values for all possible amino acid substitutions (red), GnomAD observed substitutions (blue) and observed substitutions in most constrained proteins (GnomAD LOEUF bin 0; yellow) by protein length, and **d)** distributions of Instability Heat Score for all human proteins, grouped by LOEUF functional constraint decile bins. In this boxplot, Instability heat scores are calculated with variable $\Delta\Delta G$ thresholds as a function of protein length. Thresholds were as follows (format: length_range;lower $\Delta\Delta G$ thresh; upper $\Delta\Delta G$ thresh): 1-249,-0.25,0.50;250-299,-0.25,1.00;300-349,-0.25,1.25;350-449,-0.5,1.00;450-499,-0.5,0.75;500-549,-0.75,0.5;550-599,-0.75,0.25;600-699,-0.50,0.25;700-1000,-0.25,0.25). The boxplot central bar is the median value of each respective category, the bounds of each box are the interquartile range (IQR), whiskers extend 1.5*IQR from each box and the central notches represent an approximation of the 95% confidence interval of the median box value. Boxplots of relative distributions of median B-factors for experimentally solved crystal structures grouped by functional constraint with **e)** LOEUF deciles, and **f)** S_{net} and segregated by protein lengths.